# Leveraging Modality Tags for Enhanced Cross-Modal Video Retrieval

## Abstract

Video retrieval requires aligning visual content with corresponding natural language descriptions. In this paper, we introduce Modality Auxiliary Concepts for Video Retrieval (MAC-VR), a novel approach that leverages modality-specific tags – automatically extracted from foundation models – to enhance video retrieval. Previous works have proposed to emulate human reasoning by introducing latent concepts derived from the features of a video and its corresponding caption. Building on these efforts to align latent concepts across both modalities, we propose learning auxiliary concepts from modality-specific tags. We introduce these auxiliary concepts to improve the alignment of visual and textual latent concepts, and so are able to distinguish concepts from one other. To strengthen the alignment between visual and textual latent concepts – where a set of visual concepts matches a corresponding set of textual concepts – we introduce an *Alignment Loss*. This loss aligns the proposed auxiliary concepts with the modalities' latent concepts, enhancing the model's ability to accurately match videos with their appropriate captions. We conduct extensive experiments on three diverse datasets: MSR-VTT, DiDeMo, and ActivityNet Captions. The experimental results consistently demonstrate that modality-specific tags significantly improve cross-modal alignment, achieving performance comparable to current state-of-the-art methods.

## 1 Introduction

The emergence of prominent video-sharing platforms like YouTube and TikTok has supported uploading of millions of videos daily. The demand for better video retrieval methods, which align textual queries with relevant video content, has subsequently increased. Most existing works use two main approaches. The first Fang et al. (2021); Luo et al. (2022); Jin et al. (2023b) exclusively uses word and frame features without leveraging the multi-modal information of videos. On the contrary, the second approach Dzabraev et al. (2021); Gabeur et al. (2020); Wang et al. (2021); Gabeur et al. (2022); Liu et al. (2021a); Croitoru et al. (2021) introduces additional multi-modal information from videos, such as audio, speech, objects, that are encoded and used for feature aggregation. In real-world scenarios, online videos often come with related textual information, such as tags – keywords associated with a video that describe its content and make it easier to search/filter. Few works Chen et al. (2023); Wang et al. (2022a;c) extract and exploit tags in video retrieval to better align the video and textual modalities. Inspired by these previous works, we develop a novel method called MAC-VR that integrates multi-modal information by independently extracting relevant tags **for both videos and texts**, utilizing the extensive knowledge from pre-trained Vision-Language Models (VLM) and Large Language Models (LLM) as shown in Fig.1. The example in Fig.1 shows the query "a girl doing gymnastics in the front yard", the extracted visual (VT) and textual (TT) tags include sports, physical, outdoors, and outside can help align this video to the corresponding caption. We extend the recent work DiCoSA Jin et al. (2023b), where the visual and textual coarse features are split into compact latent factors which explicitly encode visual and textual concepts. Our MAC-VR introduces visual and textual tags whose coarse features are used to learn auxiliary modality-specific latent concepts. These are aligned to latent concepts directly extracted from video and text, through an introduced Alignment Loss.

Recently, many works Liu et al. (2022); Jin et al. (2023b;c; 2022); Ibrahimi et al.; Chen et al. (2023); Wang et al. (2024a) use different inference strategies to improve the final video retrieval performance such as Querybank Normalisation (QB) Bogolin et al. (2022) and Dual Softmax (DSL) Cheng et al. (2021). In this paper, we analyse the impact of such strategies on our MAC-VR architecture to ensure

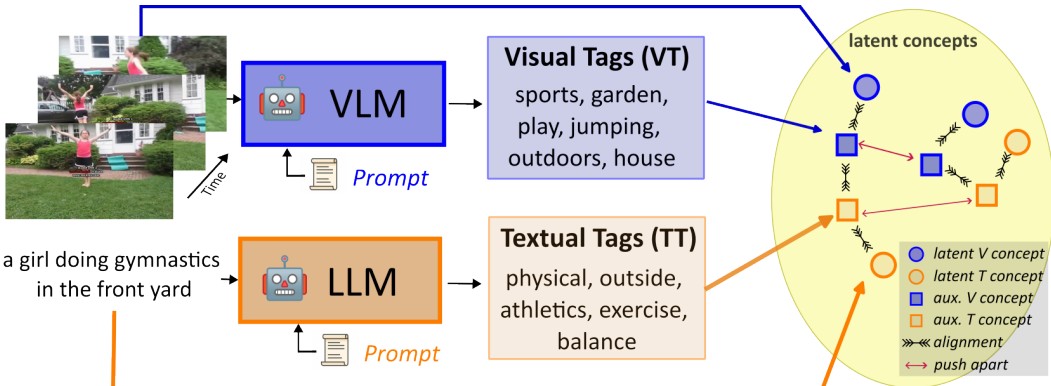

Figure 1: Tags are extracted from both videos by VLM and texts by LLM using custom prompts designed to generate the most relevant tags for each modality. For example, visual and textual tags like sports and physical can help align this video to the corresponding caption. We learn latent auxiliary concepts from these tags, that help align the videos and texts.

fair comparison with state-of-the-art (SOTA) methods. Our results show that auxiliary concepts of both modalities, in addition to the Alignment Loss, help boost the retrieval performance and better distinguish the latent concepts.

Our contribution can be summarized as follows: (i) We propose to extract modality-specific tags from foundational VLMs and LLMs to augment the video and text modalities respectively. (ii) We use these tags to learn auxiliary latent concepts in each modality to extract useful representations from the tags. (iii) We propose a new Alignment Loss to better align and distinguish these learnt latent concepts. (iv) We analyse the impact of different inference strategies on our architecture by deeply and fairly comparing our proposal with SOTA methods. (v) We conduct experiments on three datasets: MSR-VTT, DiDeMo, and ActivityNet Captions. Across all datasets, the addition of our auxiliary concepts improves performance. Detailed ablation on MSR-VTT verifies our design choices.

## 2 RELATED WORKS

**Video Retrieval.** Video retrieval aims to learn an embedding for video and text to establish effective connections between pertinent video content and natural language descriptions. Early approaches Dzabraev et al. (2021); Gabeur et al. (2020); Wang et al. (2021); Gabeur et al. (2022); Liu et al. (2021a); Croitoru et al. (2021); Fragomeni et al. (2022); Zolfaghari et al. (2021); Kunitsyn et al. (2022); Dong et al. (2022) relied on pre-trained features and/or multi-modal information inherent in videos, such as audio or speech, specialized to bridge the gap between video and text data. Notably, MMT Gabeur et al. (2020) explores multi-modal data extracted by seven pre-trained experts but integrates them without explicit guidance, employing a brute-force method. Input modalities have also been masked, e.g. in Gabeur et al. (2022), where the method can learn robust representations that enhance cross-modal matching. On the contrary, MAC-VR uses only video and text modalities without considering any additional modalities, such as audio or speech.

Recent advancements in video retrieval have followed two main methodologies. The first involves extensive pre-training of models on large-scale video-text datasets, Ge et al. (2022); Bain et al. (2021). The second focuses on transferring knowledge from image-based CLIP models Radford et al. (2021a) trained on extensive image-text pairs Kunitsyn et al. (2022); Fang et al. (2021); Gorti et al. (2022); Luo et al. (2022); Jin et al. (2023c); Huang et al. (2023); Wang et al. (2024a); Dong et al. (2023); Guan et al. (2023); Tian et al. (2024); Jin et al. (2024; 2023b); Xue et al. (2023); Fang et al. (2023). Some works Dong et al. (2023); Tian et al. (2024) use a distillation approach where a large network is first trained as a teacher network and then a smaller network is trained as a student network. In contrast, MAC-VR does not use any distillation approach and is trained directly by introducing auxiliary modality-specific tags. Similar to Jin et al. (2023b), where learnable queries and latent concepts are learnt during training, we learn latent auxiliary concepts from our modality-specific tags in addition to visual and textual latent concepts and use them as additional features to align the visual and textual concepts.

**Vision-Language and Large Language Models in Image and Video Retrieval.** The integration of Vision-Language Models (VLM) Li et al. (2023b); Liu et al. (2023a); Zhang et al. (2023b); Cheng

et al. (2024); Zhang et al. (2023a) and Large Language Models (LLM) Touvron et al. (2023a;b); Chiang et al. (2023); Dubey et al. (2024) in image Qu et al. (2024); Levy et al. (2023); Wang et al. (2024c); Zhu et al. (2024); Yan et al. (2023) and video retrieval Wu et al. (2023a); Zhao et al. (2023); Wang et al. (2022c); Shvetsova et al. (2023); Xu et al. (2024); Zhao et al. (2024); Ventura et al. (2024) has enabled significant advancements, showing an impressive understanding capabilities of these models. In Zhao et al. (2023) the authors demonstrate that LLMs can enhance the understanding and generation of video content by transferring their rich semantic knowledge. In Wu et al. (2023a; 2024), the authors explore how additional captions can enhance video retrieval by providing richer semantic context and improving matching accuracy between textual queries and video content. In contrast to these works, we do not generate additional captions as in Zhao et al. (2023); Wu et al. (2023a) but we leverage pre-trained VLMs and LLMs to generate words (i.e. visual and textual tags) that highlight relevant aspects of the action shown in the video and described by the caption.

**Tags in Image and Video understanding.** The notion of tags in image and video understanding has been previously explored in existing literature. Tags have found application across various tasks, including Video Retrieval Wang et al. (2022a); Chen et al. (2023); Wang et al. (2022c), Video Moment Retrieval Gao & Xu (2022); Wang et al. (2022b), Video Recognition Wu et al. (2023b); Kahatapitiya et al. (2024), Fashion Image Retrieval Naka et al. (2022); Wang et al. (2023a); Tian et al. (2023); Shimizu et al. (2023); Wahed et al. (2024) and Image Retrieval Huang et al. (2024); Liu et al. (2023b); Chaudhary et al. (2020); Zhu et al. (2021); Chiquier et al. (2024). Some works Wang et al. (2022a); Chen et al. (2023) use pre-trained experts to extract tags from various modalities of videos, including object, person, scene, motion, and audio. In contrast to these works, MAC-VR does not use pre-trained expert models to extract tags from a video or any additional modality such as audio. However, we generate visual and textual tags directly from videos and captions by using VLM and LLM, respectively. Similar to us, Wang et al. (2022c) uses image-language models to translate the video content into frame captions, objects, attributes, and event phrases. MAC-VR does not generate any additional caption from frames, instead using only the caption to extract tags.

## 3 MODALITY AUXILIARY CONCEPTS FOR VIDEO RETRIEVAL

We first define cross-modal text-to-video retrieval in Sec. 3.1 before describing our tag extraction approach in Sec. 3.2. Finally, in Sec. 3.3, we introduce our MAC-VR architecture.

### 3.1 CROSS-MODAL TEXT-TO-VIDEO RETRIEVAL

Given a pair $(v_i, t_i)$, where $v_i$ represents a video and $t_i$ denotes its corresponding caption, the objective of Cross-Modal Video Retrieval is to retrieve the video $v_i$ given the caption $t_i$ as query or vice versa. Typically, models use two projection functions: $f_v : v_i \longrightarrow \Omega \in \mathbb{R}^d$ and $f_t : t_i \longrightarrow \Omega \in \mathbb{R}^d$. These functions map the video and text modalities, respectively, into a shared d-dimensional latent embedding space, denoted as $\Omega$. Previous approaches aim to align the representations in this space so that the representation of a video is close to that of its corresponding caption. Following training, standard inference strategies embed a gallery of test videos and ranks these in order of their distance from each query caption. Recent approaches utilise additional inference strategies to improve performance. Two popular inference strategies are: Querybank Normalisation (QB) Bogolin et al. (2022) and Dual Softmax (DSL) Cheng et al. (2021). We introduce these here and later showcase their impact on fair comparison of the current SOTA methods.

The QB strategy was introduced to mitigate the hubness problem of high-dimensional embedding spaces Radovanovic et al. (2010), where a small subset of samples tends to appear far more frequently among the k-nearest neighbours of all embeddings. This phenomenon can have harmful effects on retrieval methods that rely on nearest-neighbour searches to identify the best matches for a given query. To mitigate this phenomenon, the similarities between embeddings are altered to minimise the influence of hubs. To do this a querybank of a set of samples is constructed from the query modality and is used as a probe to measure the hubness of the gallery. In other words, for each query, given its vector of unnormalised similarities, $S(v_j, t_i)$, over all the elements in the gallery $G$ and a probe matrix $P$, whose each row is a probe vector of similarities between the querybank and each element in the gallery, we can define a querybank normalisation function, QB, and get a vector of normalised similarities, $\eta_q = \text{QB}(S(v_j, t_i), P)$, where the querybank normalisation function is the Dynamic Inverted Softmax (DIS), introduced in Bogolin et al. (2022).

The DSL strategy is proposed to avoid one-way optimum-match in contrastive methods. DSL introduces an intrinsic prior of each pair in a batch to correct the similarity matrix and achieves the dual optimal match. In practice, we modify the original $S(v_j, t_i)$ by multiplying it with a prior $r_{i,j}$.

| Dataset | Video | Visual Tags (VT) | Textual Tags (TT) | Caption |
|---|---|---|---|---|
| MSR-VTT | | design, racing, road, driving, movement, control, speed, engine car, transportation... | product showcase, style, brand differentiation, advertising technique, automotive marketing... | a commercial for the mazda 3 the car sliding around a corner |
| DiDeMo | | birthday celebration, family gathering, cake, candle, child, birthday, making wishes, family... | event, reaction, harmony, applause, social, emotion, entertainment, collective, celebration... | the people begin to clap. |
| ActivityNet Captions | | snowy weather, tricks ride downhill, winter sports, snowboarding, outdoor activity... | hills mountains, sport, winter, outdoor, snow, challenges, fun nature, snowy terrain... | A man is seen standing on a snowy hill speaking to the camera and leads into several clips of him snowboarding... |

Figure 2: Examples of visual and textual tags (middle) for videos (left) and corresponding captions (right) across datasets.

Therefore, we can define the new similarity matrix as $\hat{S}(v_j, t_i) = r_{i,j} S(v_j, t_i)$, where the prior is defined as $r_{i,j} = \frac{exp(\tau_r S(v_i, t_i))}{\sum_j exp(\tau_r S(v_i, t_j))}$, where $\tau_r$ is a temperature hyper-parameter to smooth the gradients. While this strategy can be used both in training and inference, it is now regularly used only during inference.

## 3.2 TAG EXTRACTION

We propose to estimate tags from either the video $v_i$, using a VLM, or the text in the caption $t_i$, using an LLM. These tags are word-level representations of common objects, actions, or general ideas present in the caption or the video. They can add additional useful information to retrieve the correct video given a text query as shown in Fig. 2. For instance, given the caption: *a commercial for the Mazda 3 the car sliding around a corner*, the general tags estimated from this caption are: *product showcase, style, brand differentiation, advertising technique, automotive marketing* which reflect the commercial. These words are abstract terms that go beyond the exact caption but can help the retrieval model to better understand the specific characteristics of this caption.

In contrast, leveraging the video modality to create tags enables us to both capture a broader array of visual elements that characterise the video content and also have a representation of the video in words, facilitating matching the video content to the captions within the text modality. E.g., given the video associated with the previous caption, extracted visual tags include *road, vehicle, car, transportation, engine*, reflecting important objects in the video, and *racing, driving* reflecting the action in the video. These tags directly correspond to pertinent visual components of the video.

For extracting visual and textual tags, we use a custom prompt (see Appx. C for more details) to query the most relevant general tags for the input video $v_i$ and caption $t_i$. We extract tags individually from both modalities so they can be used for both training and inference. As the tags are extracted from a single modality, we refer to these as modality-specific tags. We detail how these tags are used within MAC-VR next.

## 3.3 ARCHITECTURE

We start from a standard Text-Conditioned Video Encoder (T-CVE) before incorporating our proposed tags into each modalities' latent concepts. These concepts are aligned and pooled to find the similarity between a video and a caption. Our proposed architecture is summarised in Fig. 3.

### 3.3.1 TEXT-CONDITIONED VIDEO ENCODER (T-CVE)

Given a caption $t_i$, we extract its text representation $T_i \in \mathbb{R}^d$. For the video representation, we first sample $N_v$ frames from a video $v_i$ and then encode them and aggregate the embedding of all frames to obtain the frame representation $F_j$ with $j \in \{1, ..., N_v\}$. Since captions often describe specific moments, as shown in previous works Bain et al. (2022); Jin et al. (2023b); Gorti et al. (2022), matching only the relevant frames improves semantic precision and reduces noise. To achieve this, we aggregate the frame representations conditioned on the text. Firstly, we calculate the inner product between the text and the frame representation $F_j$ with $j \in \{1, ..., N_v\}$:

$$a_{i,j} = \frac{exp((T_i)^\top F_j / \tau_a)}{\sum_{k=1}^{N_v} (exp((T_i)^\top F_k / \tau_a))} \quad (1)$$

where $\tau_a$ is a hyper-parameter that allows control of the textual conditioning. Then, we get the text-conditioned video representation $V_i \in R^d$ defined as $V_i = \sum_{k=1}^{N_v} a_{i,k} F_k$.

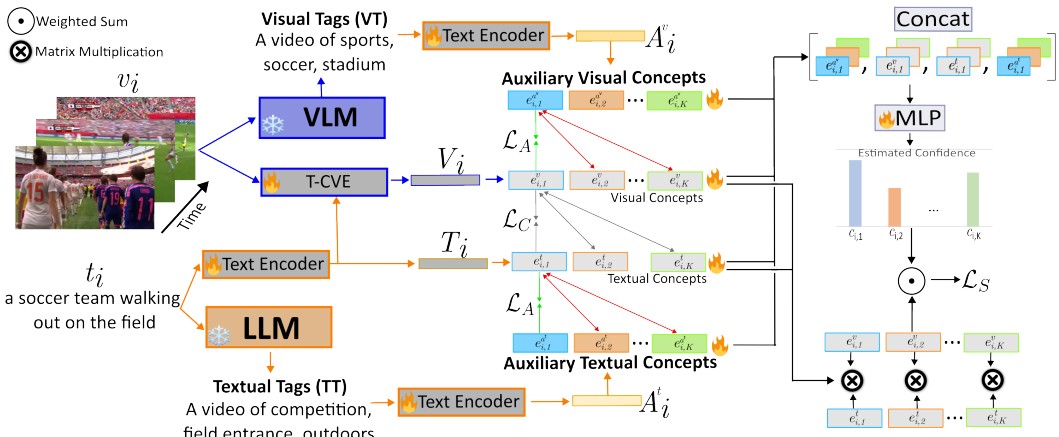

Figure 3: Architecture of MAC-VR: Given a video $v_i$ and its corresponding caption $t_i$, we generate auxiliary visual (VT) and textual (TT) tags using a VLM and an LLM, respectively. A shared text encoder projects the caption and the auxiliary tags $T_i$, $A_i^v$ and $A_i^t$ to a common space with the Text-Conditioned Video Encoder (T-CVE). Visual $e_{i,k}^v$ and textual $e_{i,k}^t$ concepts are aligned to each other by the contrastive loss $\mathcal{L}_C$ and are aligned to auxiliary visual $e_{i,k}^{a^v}$ and textual $e_{i,k}^{a^t}$ concepts by our Alignment Loss $\mathcal{L}_A$. An MLP then estimates confidence scores for each concept, to compute a weighted sum for the similarity function that is used in our Cross-Modal Loss $\mathcal{L}_S$.

### 3.3.2 LATENT CONCEPTS

To utilise both visual and textual tags in Sec 3.2 extracted from foundational models, we first randomly pick $N$ visual and textual tags during training and order them into two distinct comma-separated sentences that start with *"A video of"*. We extract visual $A_i^v$ and textual $A_i^t$ coarse tag features by using the same text encoder used for the caption. Therefore, given a video/caption pair $(v_i, t_i)$ we get a quadruple $(V_i, T_i, A_i^v, A_i^t)$. Inspired by Jin et al. (2023b), we disentangle each element of the quadruple into $K$ independent, equal-sized latent concepts. For example, when disentangling $V_i$, we get $K$ independent latent concepts, i.e. $E_i^v = [e_{i,1}^v, ..., e_{i,K}^v]$. Each latent concept $e_{i,k}^v \in R^{d/K}$ represents a distinct concept and the independence of these factors ensures that each concept is uncorrelated to the other $K-1$ latent concepts, and is thus calculated by independently projecting the text representation:

$$e_{i,k}^v = W_k^v V_i \qquad (2)$$

where $W_k^v$ is a trainable parameter. Similarly, $E_i^t$, $E_i^{a^v}$ and $E_i^{a^t}$ represent the latent concepts of the text representation $T_i$. The visual $A_i^v$ and textual $A_i^t$ tag representations are calculated in the same way. We name the $K$ latent concepts of the visual tags representation $A_i^v$ and textual tags representation $A_i^t$ as **auxiliary visual concepts** $E_i^{a^v}$ and **auxiliary textual concepts** $E_i^{a^t}$ respectively. We now have four disentangled representations for visual $E_i^v$, textual $E_i^t$, auxiliary visual $E_i^{a^v}$, and auxiliary textual $E_i^{a^t}$ concepts. Until now, these subspaces have been disentangled independently. We next describe how the alignment of these latent concepts can be used for enhancing cross-modal retrieval.

### 3.3.3 ALIGNMENT OF DISENTANGLED LATENT CONCEPTS

By default, approaches such as Jin et al. (2023b) align latent representations of videos and captions through a contrastive loss. Here, we consider aligning the auxiliary modality-specific concepts to the corresponding concepts, per modality. Specifically, we consider the visual concepts $E_i^v$ and the auxiliary visual concepts $E_i^{a^v}$. For each disentangled concept pair $(e_{i,k}^v, e_{i,k}^{a^v})$ we minimise the distance between this pair then maximise the distance to other disentangled concepts, i.e. $(e_{i,k}^v, e_{i,l}^{a^v}); l \neq k$ in a contrastive fashion to align modality concepts, similar to the loss in Jin et al. (2023b) see Appx. D for more details. Here, we focus on aligning modality concepts with our proposed auxiliary modality concepts. Recall that these latent concepts are learnt, and thus through this alignment, we aim to learn a representation of the video that matches the latent representations of the tags extracted from the VLM.

| Datasets | Visual Tags (VT) | | | | | | Textual Tags (TT) | | | | | |
| | #Tags | | | Avg #Tags | | | #Tags | | | Avg #Tags | | |
| | Train | Val | Test | Train | Val | Test | Train | Val | Test | Train | Val | Test |
|---|---|---|---|---|---|---|---|---|---|---|---|---|
| MSR-VTT Xu et al. (2016) | 63,383 | - | 12,118 | 27.69 | - | 27.83 | 320,351 | - | 8,326 | 27.17 | - | 26.52 |
| DiDeMo Hendricks et al. (2017) | 50,712 | 10,924 | 10,636 | 27.12 | 27.13 | 26.92 | 34,662 | 9,234 | 8,266 | 27.79 | 28.10 | 27.59 |
| ActivityNet Captions Krishna et al. (2017) | 58,934 | - | 35,334 | 26.83 | - | 26.79 | 29,449 | - | 21,766 | 25.17 | - | 26.05 |

Table 1: Statistical analysis of tags after extraction.

Similarly, we align the auxiliary textual concepts $E_i^{a^t}$ to the latent concepts $E_i^t$ extracted directly from the caption. We combine both modalities' alignment of latent concepts to auxiliary latent concepts and refer to this as the Alignment Loss $\mathcal{L}_A$ which aligns $E_i^v$ with $E_i^{a^v}$ and $E_i^t$ with $E_i^{a^t}$.

### 3.3.4 WEIGHTED SIMILARITY AND TRAINING LOSS

The information between the video and caption is partially matched Liu et al. (2021b). Indeed, only a subset of visual concepts are usually described in the corresponding text which might be less descriptive and informative than the video itself. Therefore, we cannot directly leverage correlations between their latent concepts, so we use adaptive pooling to define weights for the visual and textual concepts and reduce their impact on the final similarity calculation. To do this, we design an adaptive module to estimate the confidence of each cross-modal concept matching. For each concept $k$, we consider the modality and auxiliary modality concepts $[e_{i,k}^v, e_{i,k}^t, e_{i,k}^{a^v}, e_{i,k}^{a^t}]$ and use them to calculate the confidence of each cross-modal concept matching. We thus calculate:

$$c_{i,k} = MLP([e_{i,k}^v, e_{i,k}^t, e_{i,k}^{a^v}, e_{i,k}^{a^t}]) \tag{3}$$

If $c_{i,k}$ is small, the latent concept corresponding to the $k^{th}$ subspace is matched with low probability. Given this confidence, we aggregate all the visual and textual latent concept pairs to calculate the similarity of the video and text, through adaptive pooling. The similarity $S(v_i, t_i)$ is defined as:

$$S(v_i, t_i) = \sum_{k=1}^{K} c_{i,k} \frac{(e_{i,k}^t)^\top e_{i,k}^v}{\|\|e_{i,k}^t\|\|e_{i,k}^v\|} \tag{4}$$

Following common approaches, we use InfoNCE loss Gutmann & Hyvärinen (2012); Józefowicz et al. (2016) as our Cross-Modal Loss ($\mathcal{L}_S$) to optimise the cross-modal similarity $S(v_i, t_i)$, the contrastive loss $\mathcal{L}_C$ to align the modality concepts as introduced in Jin et al. (2023b) and the proposed Alignment loss $\mathcal{L}_A$ to align the modality with the auxiliary modality concepts:

$$\mathcal{L} = \mathcal{L}_S(S(v_i, t_i)) + \mathcal{L}_C(E_i^v, E_i^t) + \alpha \mathcal{L}_A(E_i^v, E_i^t, E_i^{a^v}, E_i^{a^t}). \tag{5}$$

where $\alpha$ is a weight parameter. During inference, we calculate the similarity $S(v_i, t_i)$ as in Eq. 4 for every query caption and video in the gallery. We use a fixed $M$ visual and textual tags in inference for deterministic results. Note that we do not know in this case whether the video and caption are relevant. We thus use the auxiliary modality concepts to assist in adjusting the similarity accordingly. The similarity $S(v_i, t_i)$ is then used to rank the gallery of videos for retrieval.

## 4 EXPERIMENTS

### 4.1 DATASETS AND METRICS

**MSR-VTT** Xu et al. (2016) is commonly studied in video retrieval. It comprises 10,000 videos with different content, each with 20 captions. We utilize the *9k-Train* split Gabeur et al. (2020), i.e. 9,000 videos for training and 1,000 videos for testing.

**DiDeMo** Hendricks et al. (2017) collects 10,000 Flickr videos annotated with 40,000 captions. This dataset is evaluated using a video-paragraph retrieval manner provided in Luo et al. (2022). The challenge of this dataset is to align long videos and long texts.

**ActivityNet Captions** Krishna et al. (2017) consists of 20,000 annotated YouTube videos Heilbron et al. (2015). We report results on the *val_1* split of 10,009 and 4,917 as the train and test set. We adopt the same setting in Jin et al. (2023b) to validate our model. Similar to DiDeMo, the challenge of this dataset is the alignment between long video and dense and detailed text.

**Modality-Specific Tags** We extract modality-specific tags from all videos and captions in the datasets above. Tab. 1 presents statistics of modality tags for each dataset/split.

**Metrics** We present the retrieval performance for text-to-video retrieval task using standard metrics: Recall at $L = 1, 5, 10$ ($R@L$), median rank ($MR$), and mean rank ($MeanR$).

### 4.2 IMPLEMENTATION DETAILS

We use the base code from DiCoSA Jin et al. (2023b) for our architecture. We consider this as the baseline model to which we introduce our modality tags and modality Alignment Loss.

| Method | IS | Year | MSR-VTT ($BS = 128$, $N_v = 12$) | | | | | DiDeMo ($BS = 64$, $N_v = 50$) | | | | | ActivityNet Captions ($BS = 64$, $N_v = 50$) | | | | |
|---|---|---|---|---|---|---|---|---|---|---|---|---|---|---|---|---|---|
| | | | R@1↑ | R@5↑ | R@10↑ | MR↓ | MeanR↓ | R@1↑ | R@5↑ | R@10↑ | MR↓ | MeanR↓ | R@1↑ | R@5↑ | R@10↑ | MR↓ | MeanR↓ |
| DiCoSA* Jin et al. (2023b) | - | 2023 | 47.2 | 73.5 | 83.0 | 2 | 12.9 | 41.2 | 71.3 | 81.3 | 2 | 15.9 | 36.7 | 67.8 | 81.1 | 2 | 8.7 |
| MAC-VR (ours) | - | 2024 | 48.8 | 74.4 | 83.7 | 2 | 12.3 | 43.4 | 72.5 | 82.3 | 2 | 16.9 | 37.9 | 69.4 | 81.5 | 2 | 9.6 |
| DiCoSA* Jin et al. (2023b) | QB | 2023 | 48.0 | 74.6 | 84.3 | 2 | 12.9 | 43.7 | 73.2 | 81.7 | 2 | 16.8 | 41.0 | 71.2 | 83.6 | 2 | 7.4 |
| MAC-VR (ours) | QB | 2024 | 49.3 | 75.9 | 83.5 | 2 | 12.3 | 45.5 | 74.8 | 82.3 | 2 | 16.2 | 42.4 | 73.2 | 84.1 | 2 | 8.4 |
| DiCoSA* Jin et al. (2023b) | DSL | 2023 | 52.1 | 77.3 | 85.9 | 1 | 12.9 | 47.3 | 75.7 | 83.8 | 2 | 14.2 | 44.9 | 74.8 | 85.4 | 2 | 6.8 |
| MAC-VR (ours) | DSL | 2024 | 53.2 | 77.7 | 85.3 | 1 | 10.0 | 50.2 | 76.2 | 84.2 | 1 | 15.1 | 46.5 | 75.6 | 86.2 | 2 | 6.9 |

Table 2: Comparison with baseline trained by using same training parameters of MAC-VR. * our reproduced results. IS: Inference Strategy., $BS$: Batch Size. $N_v$: Number of Frames.

| Method | IS | Year | MSR-VTT | | | | | DiDeMo | | | | | ActivityNet Captions | | | | |
|---|---|---|---|---|---|---|---|---|---|---|---|---|---|---|---|---|---|
| | | | R@1↑ | R@5↑ | R@10↑ | MR↓ | MeanR↓ | R@1↑ | R@5↑ | R@10↑ | MR↓ | MeanR↓ | R@1↑ | R@5↑ | R@10↑ | MR↓ | MeanR↓ |
| CenterCLIP Zhao et al. (2022) | - | 2022 | 48.4 | 73.8 | 82.0 | 2 | 13.8 | - | - | - | - | - | 46.2 | 77.0 | 87.6 | 2 | 5.7 |
| X-Pool Gorti et al. (2022) | - | 2022 | 46.9 | 72.8 | 82.2 | 2 | 14.3 | - | - | - | - | - | - | - | - | - | - |
| LAFF Hu et al. (2022) | - | 2022 | 45.8 | 71.5 | 82.0 | - | - | - | - | - | - | - | - | - | - | - | - |
| TS2-Net Liu et al. (2022) | - | 2022 | 47.0 | 74.5 | 83.8 | 2 | 13.0 | 41.8 | 71.6 | 82.0 | 2 | 14.8 | 41.0 | 73.6 | 84.5 | 2 | 8.4 |
| EMCL-Net Jin et al. (2022) | - | 2022 | 46.8 | 73.1 | 83.1 | 2 | - | - | - | - | - | - | 41.2 | 72.7 | - | 2 | - |
| VoP Huang et al. (2023) | - | 2023 | 44.6 | 69.9 | 80.3 | 2 | 16.3 | 46.4 | 71.9 | 81.5 | 2 | 13.6 | 35.1 | 63.7 | 77.6 | 1 | 11.4 |
| TEFAL Ibrahimi et al. | - | 2023 | 49.4 | 75.9 | 83.9 | 2 | 12.0 | - | - | - | - | - | - | - | - | - | - |
| PiDRo Guan et al. (2023) | - | 2023 | 48.2 | 74.9 | 83.3 | 2 | 12.6 | 48.6 | 75.9 | 84.4 | 2 | 11.8 | 44.9 | 74.5 | 86.1 | 2 | 6.4 |
| HBI Jin et al. (2023a) | - | 2023 | 48.6 | 74.6 | 83.4 | 2 | 12.0 | 46.9 | 74.9 | 82.7 | 2 | 12.1 | 42.2 | 73.0 | 84.6 | 2 | 6.6 |
| DiffusionRet Jin et al. (2023c) | - | 2023 | 49.0 | 75.2 | 82.7 | 2 | 12.1 | 46.7 | 74.7 | 82.7 | 2 | 14.3 | 45.8 | 75.6 | 86.3 | 2 | 6.5 |
| Prompt Switch Deng et al. (2023) | - | 2023 | 47.8 | 73.9 | 82.2 | - | 14.4 | - | - | - | - | - | - | - | - | - | - |
| Cap4Video Wu et al. (2023a) | - | 2023 | 49.3 | 74.3 | 83.8 | 2 | 12.0 | 52.0 | 79.4 | 87.5 | 1 | 10.5 | - | - | - | - | - |
| UCoFiA Wang et al. (2023b) | - | 2023 | 49.4 | 72.1 | 83.5 | 2 | 12.9 | 46.5 | 74.8 | 84.4 | 2 | 13.4 | 45.7 | 76.6 | 86.6 | 2 | 6.4 |
| PAU Li et al. (2023a) | - | 2023 | 48.5 | 72.7 | 82.5 | 2 | 13.0 | 48.6 | 76.0 | 84.5 | 2 | 12.9 | - | - | - | - | - |
| TABLE Chen et al. (2023) | - | 2023 | 47.1 | 74.3 | 82.9 | 2 | 13.4 | 47.9 | 74.0 | 82.1 | 2 | 14.3 | - | - | - | - | - |
| CLIP-ViP Xue et al. (2023) | - | 2023 | 50.1 | 74.8 | 84.6 | - | - | 48.6 | 77.1 | 84.4 | - | - | 51.1 | 78.4 | 88.3 | - | - |
| UATVR Fang et al. (2023) | - | 2023 | 47.5 | 73.9 | 83.5 | 2 | 12.3 | 43.1 | 71.8 | 82.3 | 2 | 15.1 | - | - | - | - | - |
| TeachCLIP Tian et al. (2024) | - | 2024 | 46.8 | 74.3 | - | - | - | 43.7 | 71.2 | - | - | - | 42.2 | 72.7 | - | - | - |
| MV-Adapter Jin et al. (2024) | - | 2024 | 46.2 | 73.2 | 82.7 | - | - | 44.3 | 72.1 | 80.5 | - | - | 42.9 | 74.5 | 85.7 | - | - |
| T-MASS Wang et al. (2024a) | - | 2024 | 50.2 | 75.3 | 85.1 | 1 | 11.9 | 50.9 | 77.2 | 85.3 | 1 | 12.1 | - | - | - | - | - |
| Cap4Video++ Wu et al. (2024) | - | 2024 | 50.3 | 75.8 | 85.4 | 1 | - | 52.5 | 80.0 | 87.0 | 1 | 10.3 | - | - | - | - | - |
| MAC-VR (ours) | - | 2024 | 48.8 | 74.4 | 83.7 | 2 | 12.3 | 43.4 | 72.7 | 82.3 | 2 | 16.9 | 37.9 | 69.4 | 81.5 | 2 | 9.6 |
| QB-Norm Bogolin et al. (2022) | QB | 2022 | 47.2 | 73.0 | 83.0 | 2 | - | 43.3 | 71.4 | 80.8 | 2 | - | 41.4 | 71.4 | - | 2 | - |
| DiCoSA Jin et al. (2023b) | QB | 2023 | 47.5 | 74.7 | 83.8 | 2 | 13.2 | 45.7 | 74.6 | 83.5 | 2 | 11.7 | 42.1 | 73.6 | 84.6 | 2 | 6.8 |
| DiffusionRet Jin et al. (2023c) | QB | 2023 | 48.9 | 75.2 | 83.1 | 2 | 12.1 | 48.9 | 75.5 | 83.3 | 2 | 14.1 | 48.1 | 75.6 | 85.7 | 2 | 6.8 |
| MAC-VR (ours) | QB | 2024 | 49.3 | 75.9 | 83.5 | 2 | 12.3 | 45.5 | 74.8 | 82.3 | 2 | 16.2 | 42.4 | 73.2 | 84.1 | 2 | 8.4 |
| EMCL-Net Jin et al. (2022) | DSL | 2022 | 51.6 | 78.1 | 85.3 | 1 | - | - | - | - | - | - | 50.6 | 78.9 | - | 1 | - |
| TS2-Net Liu et al. (2022) | DSL | 2022 | 51.1 | 76.9 | 85.6 | 1 | 11.7 | 47.4 | 74.1 | 82.4 | 2 | 12.9 | - | - | - | - | - |
| TEFAL Ibrahimi et al. | DSL | 2023 | 50.1 | 77.0 | 85.4 | 1 | 10.5 | - | - | - | - | - | - | - | - | - | - |
| TABLE Chen et al. (2023) | DSL | 2023 | 52.3 | 78.4 | 85.2 | 1 | 11.4 | 49.1 | 75.6 | 82.9 | 2 | 14.8 | - | - | - | - | - |
| CLIP-ViP Xue et al. (2023) | DSL | 2023 | 55.9 | 77.0 | 86.8 | - | - | 53.8 | 79.6 | 86.5 | - | - | 59.1 | 83.9 | 91.3 | - | - |
| UATVR Fang et al. (2023) | DSL | 2023 | 49.8 | 76.1 | 85.5 | 2 | 12.3 | - | - | - | - | - | - | - | - | - | - |
| T-MASS Wang et al. (2024a) | DSL | 2024 | 52.7 | 80.3 | 87.3 | 1 | 10.0 | 55.0 | 80.9 | 87.5 | 1 | 9.7 | - | - | - | - | - |
| MAC-VR (ours) | DSL | 2024 | 53.2 | 77.7 | 85.3 | 1 | 10.0 | 50.2 | 75.2 | 84.2 | 1 | 15.1 | 46.5 | 75.6 | 86.2 | 2 | 6.9 |

Table 3: Comparison with SOTA on MSR-VTT, DiDeMo and ActivityNet Captions. −: unreported results. IS: Inference Strategy.

We employ CLIP's ViT-B/32 Radford et al. (2021b) as the image encoder and CLIP's transformer base as the text encoder to encode the caption and the visual/textual tags. All encoder parameters are initialised from CLIP's pre-trained weights. We extract tags for a gallery of videos or captions in advance to decrease the computational load. For generating visual tags we utilize the fine-tuned version of VideoLLaMA2 Cheng et al. (2024) as the Vision-Language model (VLM) and Llama3.1lla (2024) as the Large Language Model (LLM) for generating textual tags. In VideoLLaMA2, Llama2 Touvron et al. (2023b) serves as a frozen LLM. We run VideoLLaMA2 by using 8 frames, sparsely sampled from the video, and different values of the temperature $\tau \in \{0.7, 0.8, 0.9, 1.0\}$. We use the same values of $\tau$ for Llama3.1. We randomly pick $N$ tags during training and we always use the first $M$ tags during inference. We concatenate single clips and single captions to get the whole video and the paragraph in DiDeMo and ActivityNet Captions to generate tags, because they are evaluated in the video-paragraph retrieval scenario. Following similar implementation and architecture details of Jin et al. (2023b), we use an Adam optimizer with linear warm-up. The initial learning rate is 1e-7 for the text encoder and video encoder and 1e-3 for other modules. Unless specified, we set $\tau_a = 3$ in Eq.1, $K = 8$ and $\alpha = 1$. Our MLP consists of two linear layers and a ReLU activation function between them, with a size of 256. The model is optimised with a batch size of 128 for MSR-VTT in 10 epochs, and a batch size of 64 in 20 epochs for DiDeMo and ActivityNet Captions. We use $N_v = 12$ frames for MSR-VTT and $N_v = 50$ frames for DiDeMo and ActivityNet Captions. We use more frames for these latter datasets because we evaluated them using a video-paragraph retrieval scenario where the whole video is considered. We use a 4-layer transformer to aggregate the embedding of all the frames. See Appx. B for some additional implementation details.

### 4.2.1 RESULTS

In Sec. 4.3, we first present the comparison of MAC-VR against the baseline DiCoSA Jin et al. (2023b) we re-ran by using our same training parameters. Then, we compare MAC-VR with SOTA methods, particularly highlighting the impact of inference strategies on fairness of comparison. Then, in Sec. 4.4, we conduct ablation studies to validate our proposal. See Appx. E for a full comparison with an additional baseline.

### 4.3 COMPARISON WITH BASELINE AND SOTA

In Tab. 2, we compare MAC-VR with our Baseline DiCoSA trained using same training parameters, more precisely the same batch size $BS$ and same number of frames $N_v$. In all the datasets we outperform our baseline DiCoSA across the three scenarios, more precisely we outperform it by $\Delta R@1 = 1.1$ for MSR-VTT, $\Delta R@1 = 2.9$ for DiDeMo and $\Delta R@1 = 1.6$ for ActivityNet Captions

| Method | R@1↑ | R@5↑ | R@10↑ | MR↓ | MeanR↓ |
|---|---|---|---|---|---|
| DiCoSA* | 52.1 | 77.3 | 85.9 | 1 | 12.9 |
| +VT | 52.1 | 77.1 | 85.3 | 1 | 11.2 |
| +$\mathcal{L}_A$ | 52.2 | 77.3 | 85.7 | 1 | 10.4 |
| +TT | 51.9 | 77.4 | 84.8 | 1 | 10.4 |
| +$\mathcal{L}_A$ | 52.0 | 77.6 | **86.0** | 1 | 10.4 |
| +VT+TT | 52.4 | 77.0 | 85.9 | 1 | 10.8 |
| +$\mathcal{L}_A$ | **53.2** | **77.7** | 85.3 | 1 | **10.0** |

| $\alpha$ | R@1↑ | R@5↑ | R@10↑ | MR↓ | MeanR↓ |
|---|---|---|---|---|---|
| 0.0 | 52.1 | 77.3 | **85.9** | 1 | 12.9 |
| 0.5 | 53.1 | 76.7 | 85.5 | **1** | **10.0** |
| 1.0 | **53.2** | **77.7** | 85.3 | **1** | **10.0** |
| 2.0 | 53.0 | 77.5 | 85.4 | 1 | 10.1 |
| 5.0 | 52.1 | 76.7 | 85.6 | 1 | 10.9 |
| 10.0 | 52.5 | 77.5 | 85.4 | 1 | 10.5 |

Table 4: Ablation on Architecture design. * our reproduced results.

Table 5: Ablation on $\alpha$ parameter of $\mathcal{L}_A$.

| Foundation Models | | R@1↑ | R@5↑ | R@10↑ | MR↓ | MeanR↓ |
|---|---|---|---|---|---|---|
| VT | TT | | | | | |
| VL | L2 | 52.0 | 77.5 | 84.9 | 1 | 10.4 |
| VL2 | L3.1 | **53.2** | **77.7** | **85.3** | **1** | **10.0** |

| Auxiliary Input | Visual | Textual | R@1↑ | R@5↑ | R@10↑ | MR↓ | MeanR↓ |
|---|---|---|---|---|---|---|---|
| Captions | Blip2 | PG | 51.2 | 76.2 | 85.0 | 1 | 11.2 |
| Captions | Blip2 | L3.1 | 51.6 | 76.7 | **85.5** | 1 | 10.9 |
| Captions | VL2 | L3.1 | 50.4 | 75.9 | 84.7 | 1 | 11.5 |
| Tags | VL2 | L3.1 | **53.2** | **77.7** | 85.3 | 1 | **10.0** |

Table 6: Ablation on foundation models. VL: Video-LLaMA. VL2: VideoLLaMA2.

Table 7: Ablation on auxiliary inputs. PG: PEGASUS. L3.1: Llama3.1. VL2: VideoLLaMA2.

when using the DSL approach as inference strategies. In general, we get the best performance on all the datasets when we use DSL as the inference strategy.

In Tab. 3, we compare MAC-VR against different SOTA works. To fairly compare MAC-VR to our baseline DiCoSA and the other SOTA methods, we consider three different settings: MAC-VR without any inference strategy, MAC-VR with QB, and MAC-VR with DSL. We split all the SOTA works based on the considered inference strategy. We get the best performance on all three datasets when we apply DSL as the inference strategy. More precisely, on MSR-VTT we outperform DiCoSA by $\Delta R@1 = 1.8$ when using QB as inference strategy and we outperform all the SOTA methods when using DSL as the inference strategy. We get comparable results with our baseline DiCoSA, when using QB, and the other SOTA methods on DiDeMo even though we use a smaller batch size $BS$ and fewer frames $N_v$. In particular, we outperform TABLE Chen et al. (2023) on MSR-VTT and DiDeMo when using the DSL strategy by $\Delta R@1 = 0.9$ and $\Delta R@1 = 1.1$, respectively. Similar to us, TABLE Chen et al. (2023) proposes to extract tags from a video by using different pre-trained experts models not only from the visual but also the audio modality. CLIP-ViP Xue et al. (2023) and T-MASS Wang et al. (2024a) outperforms MAC-VR when using DSL, but we argue that those works are not fairly comparable. CLIP-ViP Xue et al. (2023) uses a strong pre-training on WebVid-2.5M Bain et al. (2021) and HD-VILA-100M Xue et al. (2022) and T-MASS Wang et al. (2024a) proposes an inference pipeline different from ours. During inference, for each video candidate, T-MASS samples multiple stochastic text embeddings for the query text and select the closest one to the video embedding for the evaluation, using 20 sampling trials. Therefore, their results are not fairly comparable with ours: T-MASS considers additional query texts during inference whereas we consider only a single query text. Even though MAC-VR achieves lower results on ActivityNet Captions without using any inference strategy, we get comparable results with SOTA when we use QB and DSL as inference strategy. In particular, we outperform our Baseline DiCoSA by $\Delta R@1 = 0.3$ when using QB. The performance on ActivityNet Captions may be influenced by different training parameters used in other SOTA models, see Appx. E for fair comparison with SOTA. Another factor could be that we generate tags across all three datasets using the same foundation model parameters, regardless of video or caption length. ActivityNet Captions has the longest videos and captions (2 minutes per video, 50 words per paragraph), while DiDeMo has shorter videos (30 seconds, 30 words per caption) and MSR-VTT has the shortest (10-30 seconds, 10 words per caption). Using only 8 frames for long videos and concatenating captions may prevent the model from extracting detailed tags, which also explains the strong performance on the shorter MSR-VTT and DiDeMo datasets.

## 4.4 ABLATION STUDIES

All ablations are performed on the commonly used MSR-VTT dataset to validate MAC-VR.

**Number of Tags in Training and Inference.** In Fig. 4, we test our model by varying the number of visual and textual tags in training and inference. We keep the number of tags the same between the two modalities but vary that number during training and/or inference. In Fig. 4a, we adjust the number of tags in training and use the same value during inference. We show that increasing the number of tags ($> 1$) increases performance in every case. QB as an inference strategy is the least robust to changing the number of tags. When using DSL, $R@1$ increases until the best performance of $R@1 = 52.7$ with 6 tags, then the value remains stable around 52. When varying the number of tags only in inference, as shown in Fig. 4b, we observe a similar behaviour, $R@1$ increases rapidly and overcomes our baseline DiCoSA when using more than 2 tags and getting the best performance with $R@1 = 53.2$ with 8 tags. Best performance at inference is always reported at 8 tags with DSL

and 12 tags otherwise, showcasing that using more tags always increases performance. The figure also shows that DSL consistently obtains the best performance by a large margin comparing to QB or no inference strategy. We accordingly use DSL in all the remaining ablation studies.

**Architecture Design.** In Tab. 4, we ablate the components making up our proposed method MAC-VR. We first ablate the importance of modality-specific tags individually, together, and when we introduce our Alignment Loss $\mathcal{L}_A$. Using visual and textual tags individually without our Alignment Loss $\mathcal{L}_A$ gets similar results to our baseline, whereas with our Alignment Loss $\mathcal{L}_A$ improves all metrics compared to the baseline. When using both modality-specific tags simultaneously without $\mathcal{L}_A$, $R@1$ improves by $\Delta R@1 = 0.3$. Introducing our Alignment Loss $\mathcal{L}_A$, $R@1$ improves by 1.1, showcasing the importance of $\mathcal{L}_A$ in aligning the modality concepts with the corresponding auxiliary concepts.

(a) Training and Inference.  (b) Inference.

Figure 4: Ablation on varying the number of tags across all inference strategies. $(n - m)$ on the x-axis indicates the number of training tags $n$ and inference tags $m$. We vary the number of visual and textual tags in the same way.

**Parameter Alignment Loss $\mathcal{L}_A$.** The $\alpha$ parameter indicates the importance of the Alignment Loss $\mathcal{L}_A$. We consider different values of $\alpha \in \{0.0, 0.5, 1.0, 2.0, 5.0, 10.0\}$. We find that the best $R@1$ is when $\alpha = 1.0$ with $R@1 = 53.2$, and performance drops when $\alpha > 2.0$, as shown in Tab. 5.

**Choice of Foundation Models.** In Tab. 6, we ablate the use of different foundation models to extract visual and textual tags from a video and its caption. We compare Video-LLaMA Zhang et al. (2023a) against VideoLLaMA2 Cheng et al. (2024) and Llama2 Touvron et al. (2023b) against Llama3.1 lla (2024) to extract visual and textual tags. Results show that all metrics improved when using VideoLLaMA2 and Llama3.1, this is highlighted by an improvement of $\Delta R@1 = 1.2$. This can be explained by the fact that Video-LLaMA and Llama2 tend to hallucinate tags more than VideoLLaMA2 and Llama3.1, qualitative differences of the extracted tags are shown in Appx. F.

**Using auxiliary captions over tags.** In Tab. 7, we show that tags are more informative compared to new captions extracted directly from a video/paraphrased from its caption. We used different methods to extract new captions from videos using Blip2 Li et al. (2023b) and VideoLLaMA2 Cheng et al. (2024). Captions are paraphrased using PEGASUS Zhang et al. (2020) and Llama3.1 lla (2024), see Appx. G for more details. Results show that using tags outperforms other methods – specifically captions extracted by VideoLLaMA2 and Llama3.1, which are the same foundation models used to generate our tags, drops $\Delta R@1 = 2.9$.

## 5 QUALITATIVE RESULTS

Fig. 5 shows qualitative results on all three datasets. We compare the result obtained by MAC-VR against our baseline where visual and textual tags are not used. In general, both visual and textual tags add additional information extracted from the video and the caption that can help in the retrieval task. For example, consider the query *a class is being introduced to a digital reading device* and its video. Visual tags such as *student, technology* and textual tags such as *initial setup, introduction, training* add additional information. Specifically, *initial setup, introduction, training* are different words to express the main action in the video and *student, technology* add extra information: a class is comprised of *students* and a digital reading device represents *technology*.

As we introduced in Sec. 3.3.3, we align the visual and textual concepts with the corresponding auxiliary modality-specific concepts by introducing our Alignment Loss $L_A$. To show the effectiveness of our loss, we plot the t-SNE of visual and textual concepts with/without the auxiliary tags in Fig. 6, see Appx. H for additional t-SNE plots. Fig. 6a shows that without introducing visual and textual tags, some concepts are not well-separated, in particular, visual and textual concepts 7 with the textual concepts 5 and 3. This can confuse the model in the retrieval task. In contrast, by introducing the auxiliary modality-specific tags and aligning them with the corresponding visual and textual concepts, we get better-separated concepts as shown in Fig. 6b.

**Limitations and Future Works.** We find that modality-specific tags are certainly beneficial for video retrieval, but also acknowledge there are cases they are harmful. This could be either correct

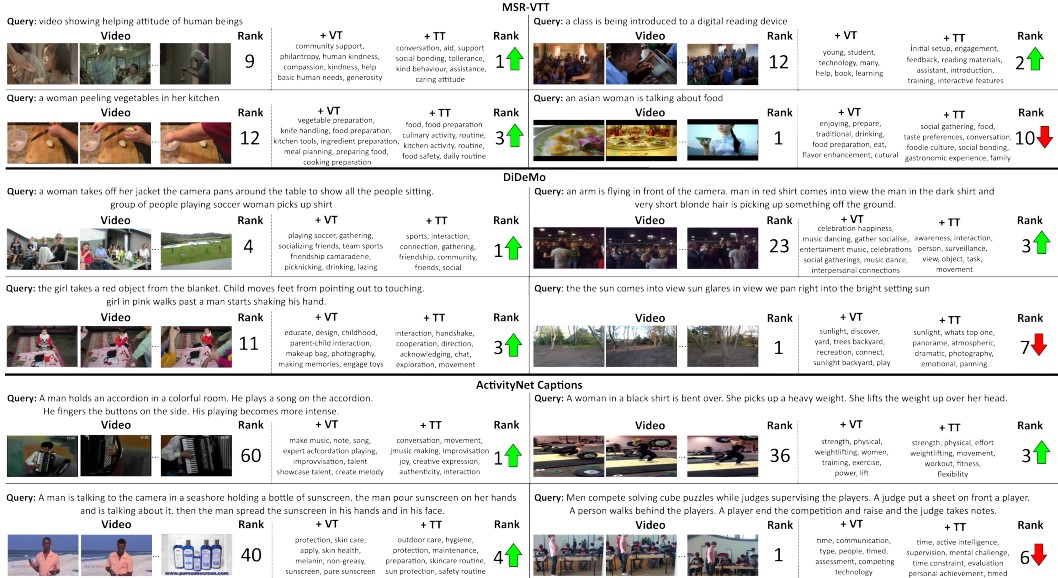

Figure 5: Qualitative results on MSR-VTT, DiDeMo and ActivityNet Captions. **Left Rank**: The ranking results of our baseline without using auxiliary tags. **Right Rank**: The ranking results of MAC-VR, which incorporates extracted visual (VT) and textual (TT) tags to enhance retrieval.

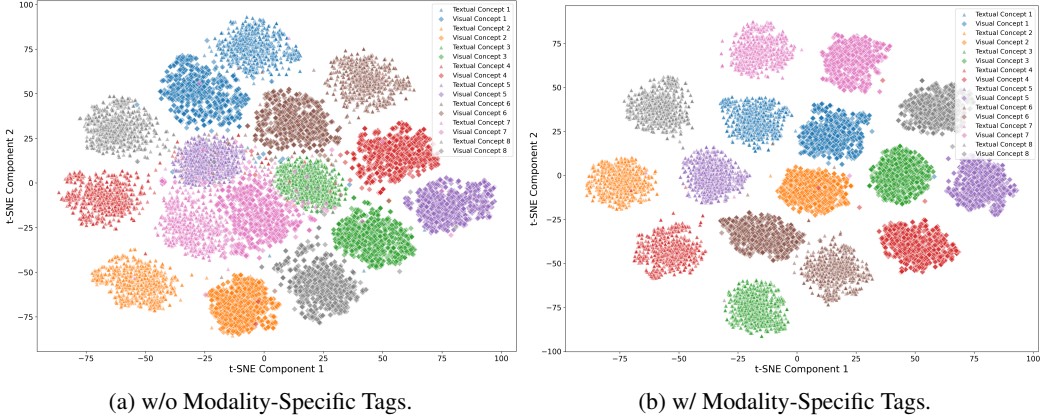

(a) w/o Modality-Specific Tags.         (b) w/ Modality-Specific Tags.

Figure 6: t-SNE plot of textual and visual concepts on the MSR-VTT test set with/without using auxiliary modality-specific tags.

tags that do not help match to the caption or incorrect tags due to errors in tag extraction. Foundation models, i.e. VLM and LLM, tend to hallucinate the content of the output meaning that the generated content might stray from factual reality or include fabricated information Rawte et al. (2023); Sahoo et al. (2024). For example, given the query *an asian woman is talking about food* we can see that *drinking* is one of the visual tags extracted. The model extracts these tags by looking at the last frame where there is a woman holding a cup and so it hallucinates the fact the woman is going to drink something. Another possible limitation of our MAC-VR is that we treat all the tags with the same importance. It is possible that some generated words can be more common than others and therefore less discriminative. We leave this for future work. See Appx. I for more details on the reported limitations.

# 6 CONCLUSION

In this work, we introduce the notion of visual and textual tags extracted by foundation models from a video and its caption respectively and use them to boost the video retrieval performance. We propose MAC-VR (Modality Auxiliary Concepts for Video Retrieval), where we incorporate modality-specific auxiliary tags, projected into disentangled auxiliary concepts. We use a new Alignment Loss to better align each modality with its auxiliary concepts. We ablate our method to further show the benefit of using auxiliary modality-specific tags in video retrieval. Our results indicate, both qualitatively and by comparing to other approaches, that modality-specific tags help to decrease ambiguity in video retrieval on three video datasets.

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

## A    APPENDIX CONTENTS

In the Appendix, we present further information about MAC-VR and ablate our design choices. In Appx. B we add some additional implementation details of MAC-VR. Within Appx. C, we present the prompts used to extract tags from the foundation models. Next, in Appx. D we explain better the contrastive loss $\mathcal{L}_C$ and the proposed Alignment loss $\mathcal{L}_A$. In Appx. E, we provide further comparison with an additional baseline in which tags are appended to the main caption and with some SOTA trained by using our same training parameters. After, we showcase the effect of using tags extracted from different foundation models in Appx. F. We present a comparison to auxiliary captions in Appx. G, further t-SNE plots of MAC-VR in Appx. H, and finally discuss limitations of MAC-VR in Appx. I.

## B    ADDITIONAL IMPLEMENTATION DETAILS

In Sec. 4.2 we described the implementation details of MAC-VR. To get all the results shown in Tab.2 and Tab. 3 we used 12 tags both in training and inference when evaluating MAC-VR without any inference strategy and with the QB, and 6 and 8 tags in training and inference respectively when using the DSL.

## C    TAG EXTRACTION

In Sec. 3.2, we have described how we extracted tags from a video and its corresponding caption, here we provide the prompts used to extract tags for the video and text modalities. The prompt used as input of VideoLLaMA2 Cheng et al. (2024) is:

> A general tag of an action is a fundamental and overarching idea that encapsulates the essential principles, commonalities, or recurrent patterns within a specific behavior or activity, providing a higher-level understanding of the underlying themes and purpose associated with that action.
> What are the top 10 general tags that capture the fundamental idea of this action? Give me a bullet list as output where each point is a general tag, and use one or two significant words per tag and do not give any explanation.

and the prompt of Llama3.1 lla (2024) is:

> A chat between a curious user and an artificial intelligence assistant. The assistant gives helpful, detailed, and polite answers to the user's questions.
> USER: You are a conversational AI agent. You typically extract general tags of an action.
>
> A general tag of an action is a fundamental and overarching idea that encapsulates the essential principles, commonalities, or recurrent patterns within a specific behavior or activity, providing a higher-level understanding of the underlying themes and purpose associated with that action.
>
> Given the following action: 1) {}
>
> What are the top 10 general tags of the above action? Use one or two significant words per tag and do not give any explanation.
>
> ASSISTANT:

Note that {} will be replaced with the caption. Even though we have not provided any example in the prompts, the foundation models have been able to generate reasonable outputs for both video and text. We do not use any strategy to avoid the hallucination problem of foundation models as the results were found via spot checking to be clean enough for our purposes. The only post-processing strategy we adopted was to clean the output of the models in order to get the corresponding tags: we remove punctuation; stopwords; extracted tags that contain a noun and a verb to avoid the presence of complete sentences as tags; and tags larger than 3 words. In Fig.7 we show additional examples of tags for all the three datasets.

## D    ALIGNMENT OF DISENTANGLED LATENT CONCEPTS

The $L_C$ proposed in Jin et al. (2023b) to align latent representations of videos and captions consists of two terms: an Inter-Concept Decoupling and an Intra-Concept Alignment term. The *Inter-Concept Decoupling* loss aims to ensure that latent subspaces capturing different semantic aspects

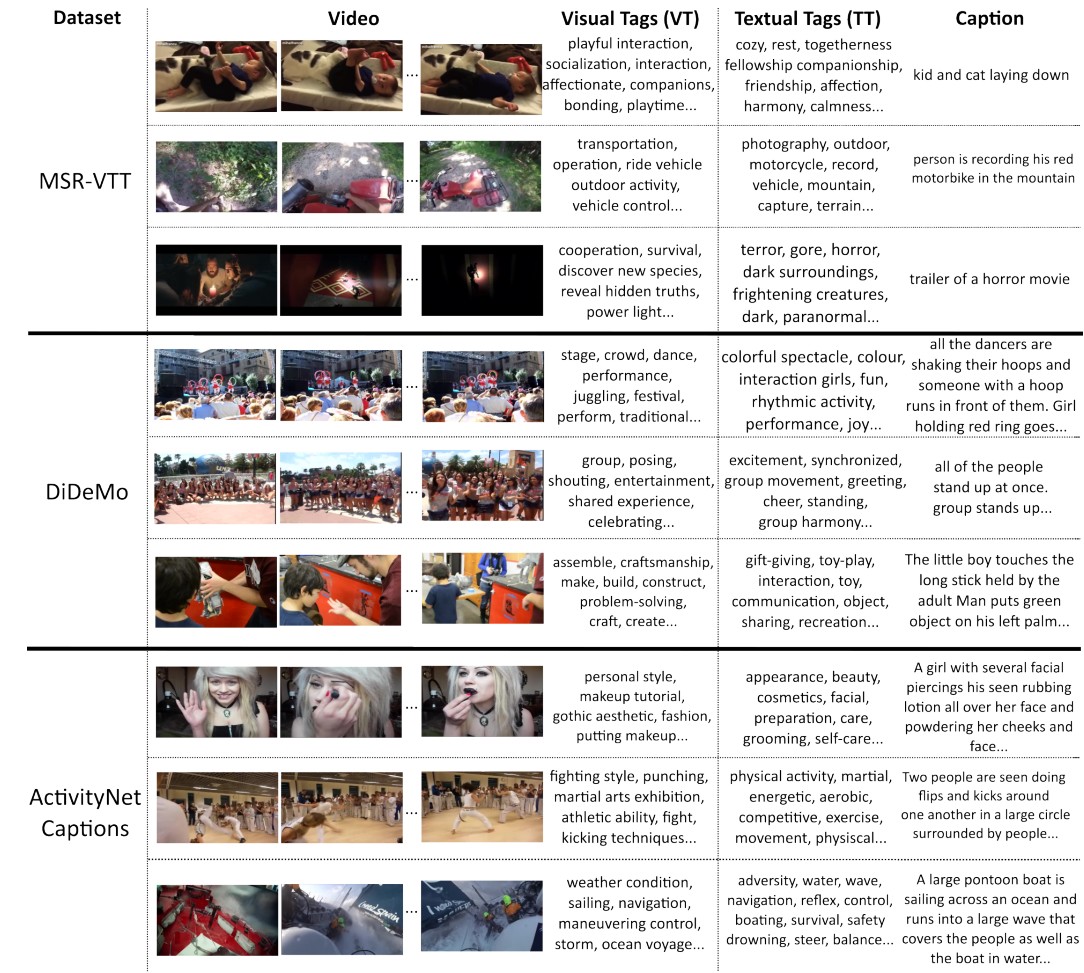

Figure 7: Additional examples of visual and textual tags across our datasets.

of text and video representations are minimally correlated. This separation allows each subspace to focus on unique semantic features without overlap, enhancing the overall discriminative power.

To do so, the mutual information between latent factors is used to quantify their dependency. Given the two latent concepts $e_{i,k}^t$ and $e_{i,l}^v$, their mutual information is defined in terms of their probabilistic density functions:

$$I(e_{i,k}^t; e_{i,l}^v) = \mathbb{E}_{\mathbf{t},\mathbf{v}}[p(e_{i,k}^t, e_{i,l}^v) log \frac{p(e_{i,k}^t, e_{i,l}^v)}{p(e_{i,k}^t) \cdot p(e_{i,l}^v)}] \tag{6}$$

However, since direct computation is challenging, the covariance $C_{k,l}$ between normalized latent factors is used as a proxy. By minimizing this covariance for unrelated subspaces through a loss function:

$$L_1 = \sum_k \sum_{l \neq k} (C_{k,l})^2 \tag{7}$$

the model effectively reduces inter-concept mutual dependencies, achieving conceptual disentanglement. The *Intra-Concept Alignment* loss focuses on strengthening the correspondence between text and video representations within the same semantic subspaces. By maximizing mutual information between positive pairs, the alignment ensures that corresponding subspaces align semantically. This is implemented through a loss:

$$L_2 = \sum_k (1 - C_{k,k})^2 \tag{8}$$

which encourages high covariance for aligned pairs, ensuring that the semantic alignment within subspaces is robust and accurate. The combination of these two approaches is captured in the total

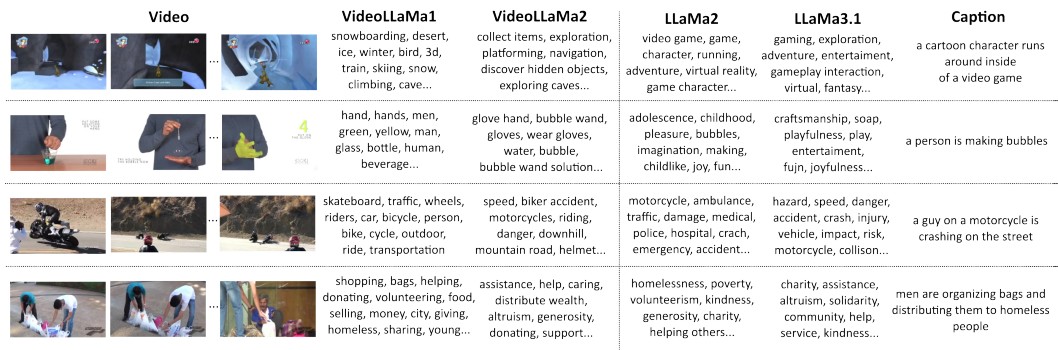

Figure 8: Comparison of extracted visual and textual tags on the MSR-VTT dataset when using different foundation models.

loss function

$$L_C = \gamma L1 + \delta L_2 \qquad (9)$$

where $\gamma$ and $\delta$ are weights to balance the importance of decoupling and alignment. Our Alignment Loss $\mathcal{L}_A$ has the same formulation as explained above and we use the same weights already ablated in Jin et al. (2023b).

## E  COMPARISON WITH ADDITIONAL BASELINE AND SOTA

In Tab. 2, we have shown the comparison of MAC-VR against our main baseline DiCoSA Jin et al. (2023b). In Tab. 8, we show the complete comparison with our baselines, where we define an additional baseline DiCoSA-ext that is an extension of DiCoSA Jin et al. (2023b) where we append the extracted tags at the end of the original caption of each video only during training. As we can see from the results, adding tags at the end of the caption does not help to learn better visual and textual concepts, indeed DiCoSA-ext performs even worse than the original DiCoSA. Without additional modelling capacity, the model struggles to benefit from the additional information.

In Tab.3, we have shown the comparison of MAC-VR against SOTA methods. As we explained in Sec.4.3, many SOTA works use different training parameters when training on DiDemo and ActivityNet Captions, more precisely different values of batch size ($BS$) and number of frames ($N_v$), such as EMCL-Net Jin et al. (2022) and UCoFiA Wang et al. (2023b). We re-run these methods by using our same training parameters and show the results in Tab.9. MAC-VR outperforms UCoFIA on DiDeMo and using our same training parameters the performance on ActivityNet Captions of UCoFIA drops drastically getting closer to our performance. Similarly, MAC-VR outperforms EMCL-NeT on DIDeMo and gets very similar performance on ActivityNet Captions.

## F  DIFFERENCES BETWEEN TAGS EXTRACTED FROM DIFFERENT FOUNDATIONS MODELS

As explained in Sec. 4.4, we considered different foundation models to extract visual and textual tags. More precisely we considered Video-LLaMA Zhang et al. (2023a) and VideoLLaMA2 Cheng et al. (2024) to extract visual tags and Llama2 Touvron et al. (2023b) and Llama3.1 lla (2024) to extract textual tags. We used the same parameters and same prompts to extract the tags by using all the considered foundations models. As shown in Tab. 10, the total number of extracted textual tags by Llama2 Touvron et al. (2023b) is much smaller than the total number obtained when using Llama3.1 lla (2024). Same conclusion for the total number of extracted visual tags by using Video-LLaMA Zhang et al. (2023a) and VideoLLaMA2 Cheng et al. (2024) as shown in Tab. 11. This show a less ability of these foundation models to generate more tags compared with the more recent ones. In particular, this is more evident when we focused on the visual tags, where not only the total number of unique tags is smaller but also the average number of tags per pairs is the same one, meaning that many tags are shared among pairs and so there are less unique tags able to distinguish all the pairs.

Fig. 8 shows a qualitative comparison of the extracted tags by using different foundations models. It is evident that Video-LLaMA and Llama2 tend to hallucinate tags that are not relevant with what shown in the video and described in the caption. Moreover, the textual tags extracted by using Llama2 are very often words that already appear in the caption. For example, given the captions *a*

| Method | IS | Year | MSR-VTT ($BS=128, N_v=12$) | | | | | DiDeMo ($BS=64, N_v=50$) | | | | | ActivityNet Captions ($BS=64, N_v=50$) | | | | |
|---|---|---|---|---|---|---|---|---|---|---|---|---|---|---|---|---|---|
| | | | R@1↑ | R@5↑ | R@10↑ | MR↓ | MeanR↓ | R@1↑ | R@5↑ | R@10↑ | MR↓ | MeanR↓ | R@1↑ | R@5↑ | R@10↑ | MR↓ | MeanR↓ |
| DiCoSA* Jin et al. (2023b) | - | 2023 | 47.2 | 73.5 | 83.0 | 2 | 12.9 | 41.2 | 71.3 | 81.3 | 2 | 15.9 | 36.7 | 67.8 | 81.1 | 2 | 8.7 |
| DiCoSA-ext | - | 2024 | 42.1 | 69.0 | 77.7 | 2 | 18.1 | 37.7 | 70.1 | 79.5 | 2 | 20.0 | 36.1 | 67.4 | 80.3 | 3 | 9.4 |
| **MAC-VR (ours)** | - | 2024 | **48.8** | **74.4** | **83.7** | 2 | **12.3** | **43.4** | **72.5** | **82.3** | 2 | 16.9 | **37.9** | **69.4** | **81.5** | 2 | 9.6 |
| DiCoSA* Jin et al. (2023b) | QB | 2023 | 48.0 | 74.6 | **84.3** | 2 | 12.9 | 43.7 | 73.2 | 81.7 | 2 | 16.8 | 41.0 | 71.2 | 83.6 | 2 | **7.4** |
| DiCoSA-ext | QB | 2024 | 45.3 | 69.5 | 79.1 | 2 | 17.1 | 41.5 | 71.9 | 81.2 | 2 | 18.9 | 40.0 | 70.5 | 82.7 | 2 | 8.1 |
| **MAC-VR (ours)** | QB | 2024 | **49.3** | **75.9** | 83.5 | 2 | **12.3** | **45.5** | **74.8** | **82.3** | 2 | **16.2** | **42.4** | **73.2** | **84.1** | 2 | 8.4 |
| DiCoSA* Jin et al. (2023b) | DSL | 2023 | 52.1 | 77.3 | **85.9** | 1 | 12.9 | 47.3 | 75.7 | 83.8 | 2 | 14.2 | 44.9 | 74.8 | 85.4 | 2 | **6.8** |
| DiCoSA-ext | DSL | 2024 | 50.2 | 74.5 | 84.4 | 1 | 12.4 | 47.0 | 74.7 | 81.6 | 2 | 15.6 | 44.7 | 74.2 | 85.1 | 2 | 7.3 |
| **MAC-VR (ours)** | DSL | 2024 | **53.2** | **77.7** | 85.3 | 1 | **10.0** | **50.2** | **76.2** | **84.2** | 1 | 15.1 | **46.5** | **76.2** | **86.2** | 2 | 6.9 |

Table 8: Full comparison with additional baseline trained by using same training parameters of MAC-VR. * our reproduced results. IS: Inference Strategy. $BS$: Batch Size. $N_v$: Number of Frames.

| Method | IS | Year | DiDeMo ($BS=64, N_v=50$) | | | | | ActivityNet Captions ($BS=64, N_v=50$) | | | | |
|---|---|---|---|---|---|---|---|---|---|---|---|---|
| | | | R@1↑ | R@5↑ | R@10↑ | MR↓ | MeanR↓ | R@1↑ | R@5↑ | R@10↑ | MR↓ | MeanR↓ |
| UCoFiA Wang et al. (2023b) | - | 2023 | 42.1 | 69.2 | 79.1 | **2** | **16.3** | **41.2** | **73.6** | **84.3** | 2 | **7.9** |
| **MAC-VR (ours)** | - | 2024 | **43.4** | **72.5** | **82.3** | 2 | 16.9 | 37.9 | 69.4 | 81.5 | 2 | 9.6 |
| EMCL-Net Jin et al. (2022) | DSL | 2022 | 47.6 | 73.5 | 82.8 | 2 | **11.9** | **47.1** | **75.7** | **86.4** | 2 | 7.0 |
| **MAC-VR (ours)** | DSL | 2024 | **50.2** | **76.2** | **84.2** | 1 | 15.1 | 46.5 | 75.6 | 86.2 | 2 | **6.9** |

Table 9: Comparison with Baseline trained by using same training parameters of MAC-VR. * our reproduced results. IS: Inference Strategy., BS: Batch Size. $N_v$: Number of Frames.

| Datasets | Textual Tags | | | | | | | | | | | |
|---|---|---|---|---|---|---|---|---|---|---|---|---|
| | LLaMa2 | | | | | | LLaMa3.1 | | | | | |
| | #Tags | | | Avg #Tags | | | #Tags | | | Avg #Tags | | |
| | Train | Val | Test | Train | Val | Test | Train | Val | Test | Train | Val | Test |
| MSR-VTT Xu et al. (2016) | 162,571 | - | 5,058 | 14.96 | - | 15.20 | 320,351 | - | 8,326 | 27.17 | - | 26.52 |
| DiDeMo Hendricks et al. (2017) | 19,208 | 4,537 | 4,332 | 13.02 | 13.00 | 13.19 | 34,662 | 9,234 | 8,266 | 27.79 | 28.10 | 27.59 |
| ActivityNet Captions Krishna et al. (2017) | 23,500 | - | 14,576 | 15.97 | - | 16.05 | 29,449 | - | 21,766 | 25.17 | - | 26.05 |

Table 10: Comparison of statistics of textual tags on the MSR-VTT dataset when using different foundation models.

| Datasets | Visual Tags | | | | | | | | | | | |
|---|---|---|---|---|---|---|---|---|---|---|---|---|
| | Video-LLaMA | | | | | | VideoLLaMA2 | | | | | |
| | #Tags | | | Avg #Tags | | | #Tags | | | Avg #Tags | | |
| | Train | Val | Test | Train | Val | Test | Train | Val | Test | Train | Val | Test |
| MSR-VTT Xu et al. (2016) | 8,049 | - | 3,500 | 27.11 | - | 27.12 | 63,383 | - | 12,118 | 27.69 | - | 27.83 |
| DiDeMo Hendricks et al. (2017) | 21,204 | 6,103 | 5,743 | 31.5 | 31.20 | 31.21 | 50,712 | 10,924 | 10,636 | 27.12 | 27.13 | 26.92 |
| ActivityNet Captions Krishna et al. (2017) | 18.738 | - | 12,409 | 27.35 | - | 27.23 | 58,934 | - | 35,334 | 26.83 | - | 26.79 |

Table 11: Comparison of statistics of visual tags on the MSR-VTT dataset when using different foundation models.

*cartoon character runs around inside of a video game* and its corresponding video, we can see that Video-LLaMA and Llama2 hallucinate some visual tags such as *snowboarding, desert, skiing, bird, climbing*—definitely irrelevant to what appear in the video—and textual tags such as *video game, running, character, game character* that are already words that appear in the caption, therefore they do not add any additional information to better retrieve the correct video. On the contrary VideoLLaMA2 and Llama3.1 tend to extract tags that add additional information to the video and text. See Fig. 8 for more example on all the considered datasets.

# G HOW TO GENERATE AUXILIARY CAPTIONS.

In Sec. 4.4, we ablate the use of auxiliary captions instead of using tags. We generate these additional captions by extracting them directly from the video and paraphrasing the original caption. We consider different approaches to extract captions from video and text.

**Visual Captions.** We consider two different approaches to generate new captions from a video: Blip2 Li et al. (2023b) and VideoLLaMA2 Cheng et al. (2024). Following the same approach proposed in Wang et al. (2024b), we generate new captions by extracting the middle frame of each video and use Blip2 to generate a new caption. On the contrary, we used a general prompt to ask VideoLLaMA2 to generate new captions. The parameters of VideoLLaMA2 are the same ones we used to extract visual tags in MAC-VR, as described in Sec. 4.2.

> You are a conversational AI agent. You typically look at a video and generate a new caption for a video. Generate 10 new captions. Give me a bullet list as output.

**Textual Captions.** We consider the paraphraser PEGASUS Zhang et al. (2020) and Llama3.1 lla (2024) to paraphrase the original caption. PEGASUS Zhang et al. (2020) is a standard Transformer-based encoder-decoder method pre-trained on a massive text corpora with a novel pre-training objective called Gap Sentence Generation (GSG). Instead of using traditional language modeling, PEGASUS removes important sentences from a document (gap-sentences) and asks the model to predict these missing sentences. After the pre-training stage, PEGASUS is fine-tuned on specific summarization datasets to improve its performance on downstream tasks. The model becomes highly effective at generating concise and accurate summaries by leveraging its pre-training knowledge.

We extract new captions from a caption by Llama3.1 lla (2024) by giving as input a general prompt as we did to extract tags:

> A chat between a curious user and an artificial intelligence assistant. The assistant gives helpful, detailed, and polite answers to the user's questions. USER: You are a conversational AI agent. You typically paraphrase sentences by using different words but keeping the same meaning.
>
> Given the following sentence: 1) {}
>
> Generate 10 different sentences that are a paraphrased version of the original sentence. Give me a bullet list as output.
>
> ASSISTANT:

We do not apply any strategy to avoid the hallucination problem as we did to extract tags.
We randomly pick an extracted visual and textual caption as auxiliary inputs in MAC-VR during training. In inference, we always pick the first caption in the set of the extracted ones. Some examples of the extracted captions with all the considered methods are shown in Fig. 9.

## H  ADDITIONAL T-SNE PLOT ON MSR-VTT, DIDEMO AND ACTIVITYNET CAPTIONS.

In Fig. 10, we show the t-SNE plot of MAC-VR on the MSR-VTT test set when using modality-specific tags with/without our Alignment loss $\mathcal{L}_A$. As we can see, the use of $\mathcal{L}_A$ helps to better distinguish the different concepts and have better clusters in the t-SNE plot. In Fig. 11 and Fig. 12, we show the t-SNE plot of visual and textual concepts without auxiliary modality-specific tags and when using only visual tags (Fig. 11) and textual tags (Fig. 12) with our Alignment loss $\mathcal{L}_A$. As we can see, both tags used individually help to better align the visual and textual concepts, in particular the visual tags helps to better align the visual and textual concepts compared to the textual tags. A possible explanation is that tags extracted from videos share the same modality as captions, which facilitates better alignment between visual and textual concepts. We leave this conclusion as possible inspiration for future works in this field. In Fig. 13 and Fig.14, we show the t-SNE plot of MAC-VR on the MSR-VTT test set of visual and textual concepts with/without auxiliary modality-specific tags similar to what we did in Sec. 5 for MSR-VTT. We can see that the use of auxiliary modality-specific tags help to better distinguish the different concepts and have better clusters in the t-SNE plot. This behave is more evident in DiDeMo (i.e. Fig. 13) rather than ActivityNet Captions (i.e. Fig. 14). A possible explanation might be the fact that the captions are longer than the MSR-VTT and so already include more information that can be used to better distinguish the visual and textual modality concepts.

## I  LIMITATIONS OF MAC-VR

As mentioned in Sec. 5, two possible limitation of MAC-VR might be the hallucination problem of foundation models and the long-tailed distribution of tags.
**Hallucination Problem.** Fig. 15 show some additional examples where MAC-VR fails. A general problem that is evident from these examples is that the model sometimes tend to extract wrong tags that are not related with what is shown in the video or described in the caption.
For example, some visual and textual tags of the video associated to the caption *a man and woman performing in front of judes* are *musical performance, thematic music* as visual tags and *law, testimony, marriage, couple court*. These tags are not relevant to what shown in the video and described in the text. The model hallucinates textual tags as *law, testimony, marriage, couple court* because in the caption there are words such as *man, woman. judes* that can be associated wrongly with the extracted *law, testimony, marriage, couple court*, and visual tags such as *musical performance, thematic music* because there are people performing something on stage so the most common association might be a musical performance. See Fig. 15 to see other examples.
**Long-tailed Distribution of Tags.** Fig. 16 to 29 show the distribution of the top-250 visual and textual tags in training and testing for all the three datasets. In general we can see that the distribution of these tags is long-tailed and there are some tags that are very common. Consequently, the most common tags are shared among many pairs in the dataset, but we find that combinations of tags are still unique enough to provide discriminative information for the model.

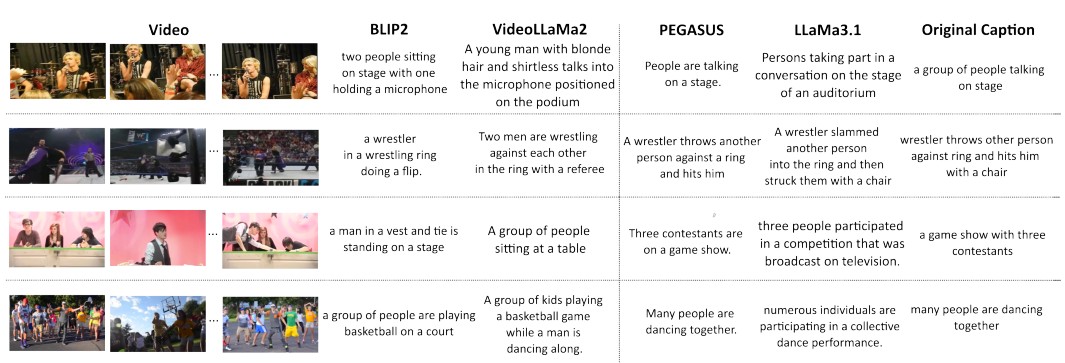

Figure 9: Examples of extracted captions on the MSR-VTT dataset.

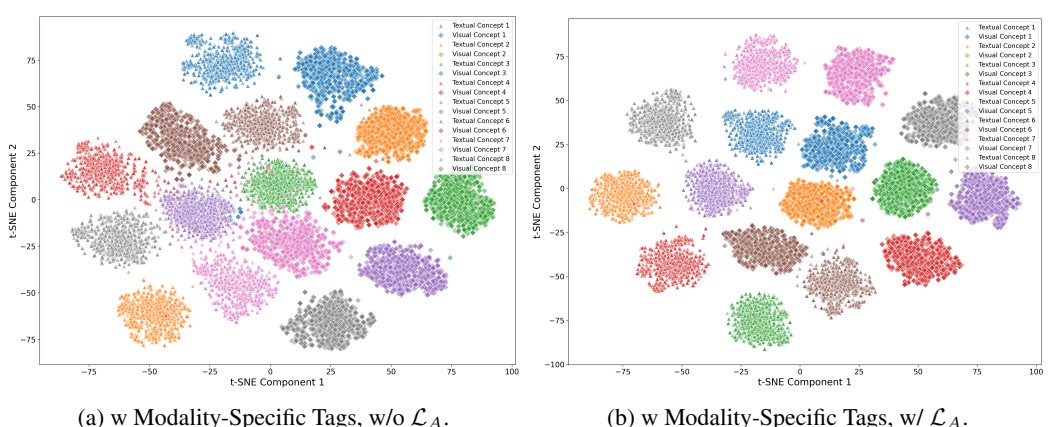

(a) w Modality-Specific Tags, w/o $\mathcal{L}_A$.    (b) w Modality-Specific Tags, w/ $\mathcal{L}_A$.

Figure 10: t-SNE plot of visual and textual concepts on the MSR-VTT test set with/without the Alignment Loss $\mathcal{L}_A$ with using both visual and textual tags.

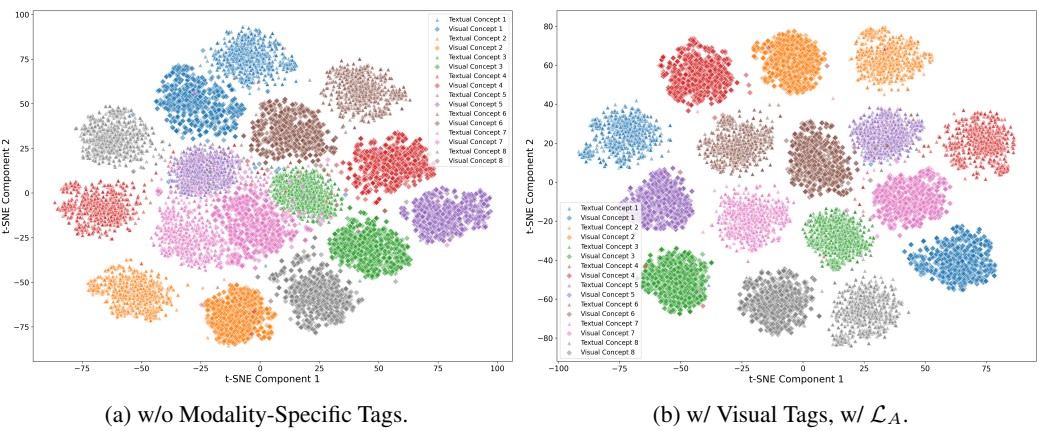

(a) w/o Modality-Specific Tags.    (b) w/ Visual Tags, w/ $\mathcal{L}_A$.

Figure 11: t-SNE plot of visual and textual concepts on the MSR-VTT test set without using auxiliary modality-specific tags and with using only visual tags.

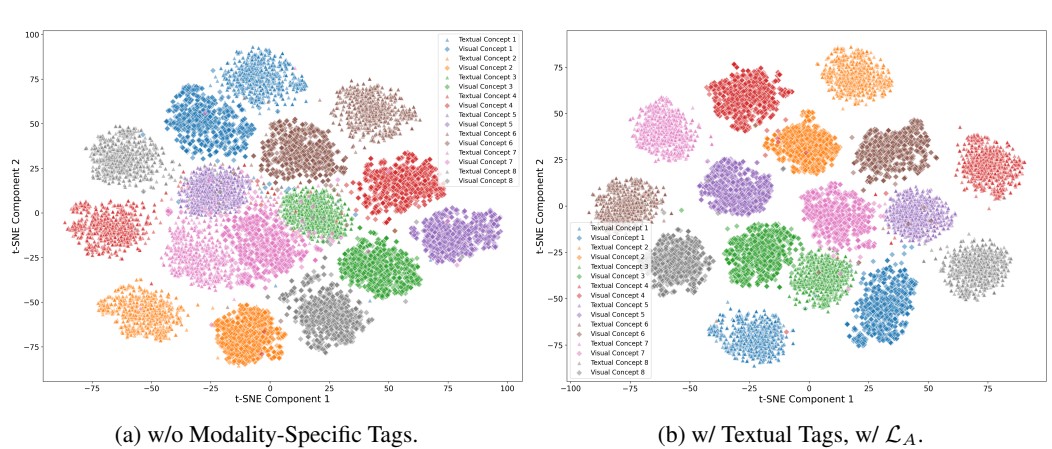

(a) w/o Modality-Specific Tags.

(b) w/ Textual Tags, w/ $\mathcal{L}_A$.

Figure 12: t-SNE plot of visual and textual concepts on the MSR-VTT test set without using auxiliary modality-specific tags and with using only textual tags.

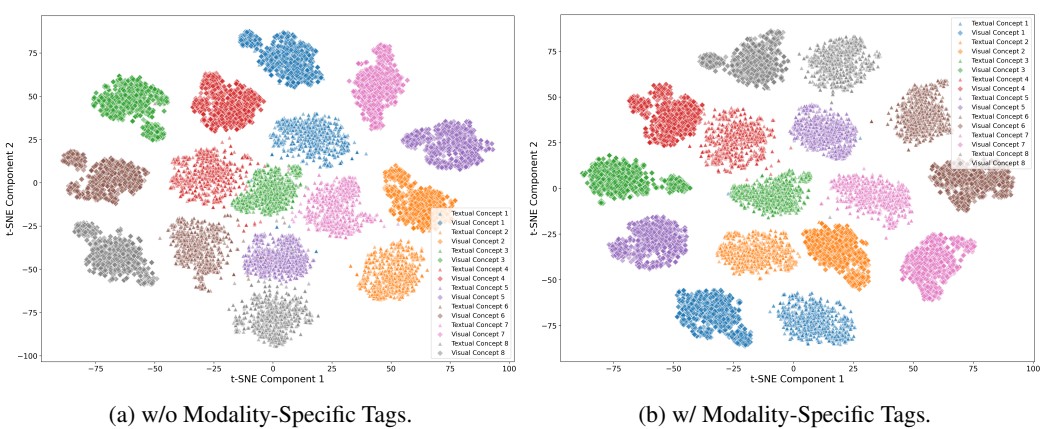

(a) w/o Modality-Specific Tags.

(b) w/ Modality-Specific Tags.

Figure 13: t-SNE plot of visual and textual concepts on the DiDeMo test set with/without using auxiliary modality-specific tags.

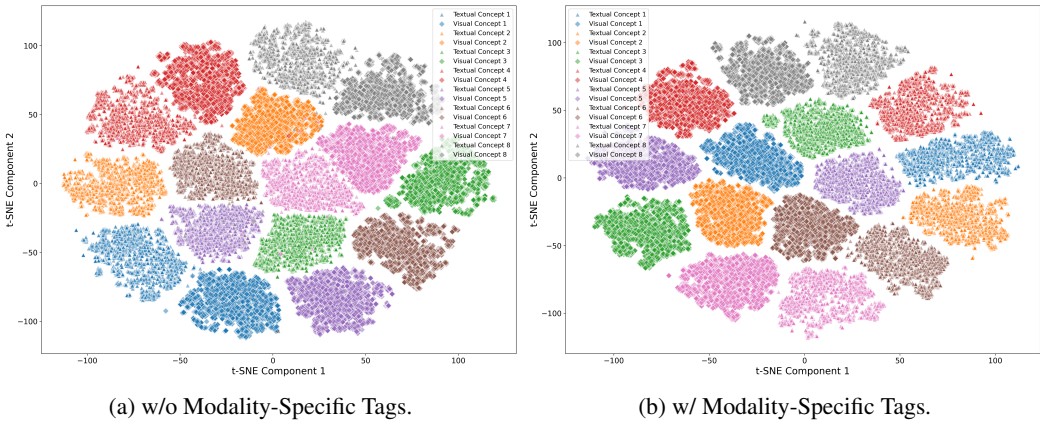

(a) w/o Modality-Specific Tags.

(b) w/ Modality-Specific Tags.

Figure 14: t-SNE plot of visual and textual concepts on the ActivityNet Captions test set with/without using auxiliary modality-specific tags.

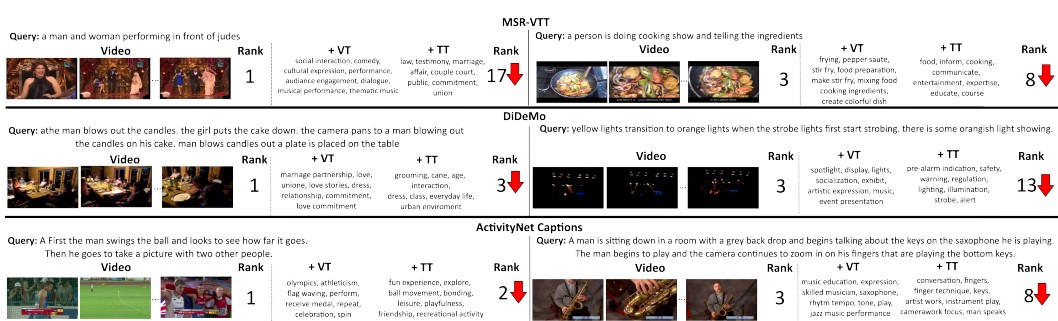

Figure 15: Additional failure cases of MAC-VRacross our datasets.

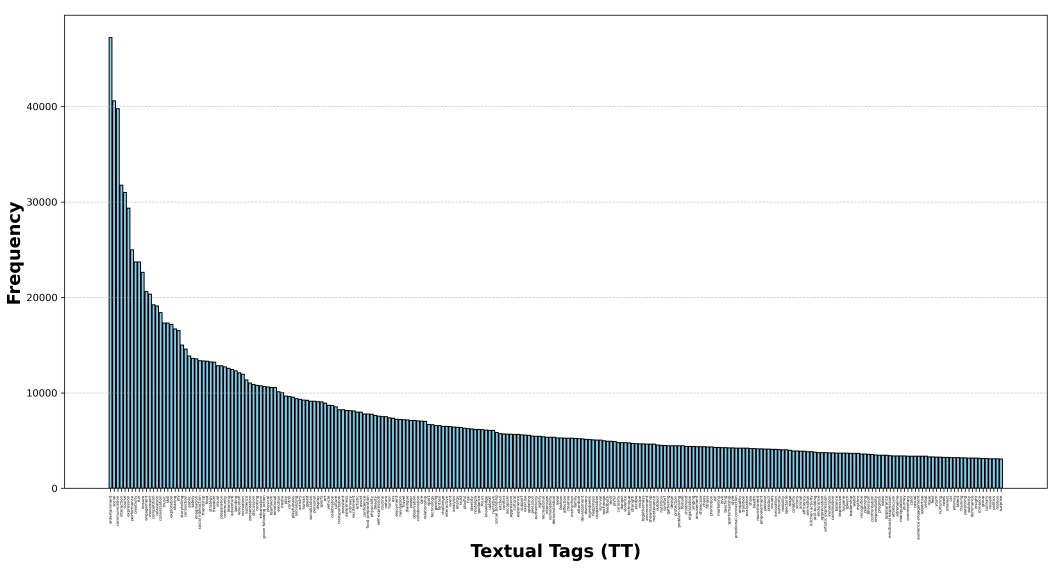

Figure 16: Distribution of the top-250 training textual tags of MSR-VTT.

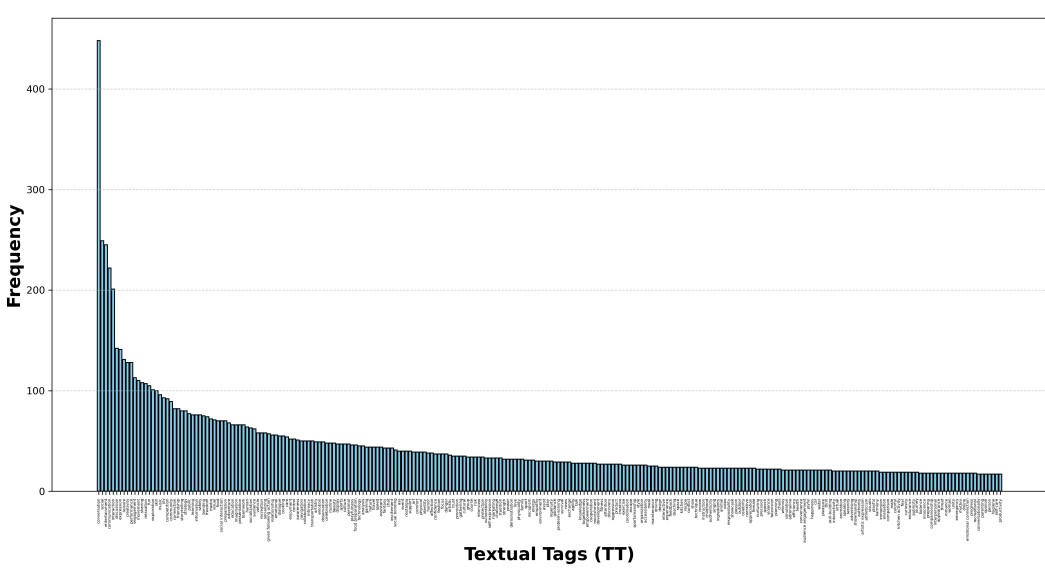

Figure 17: Distribution of the top-250 testing textual tags of MSR-VTT.

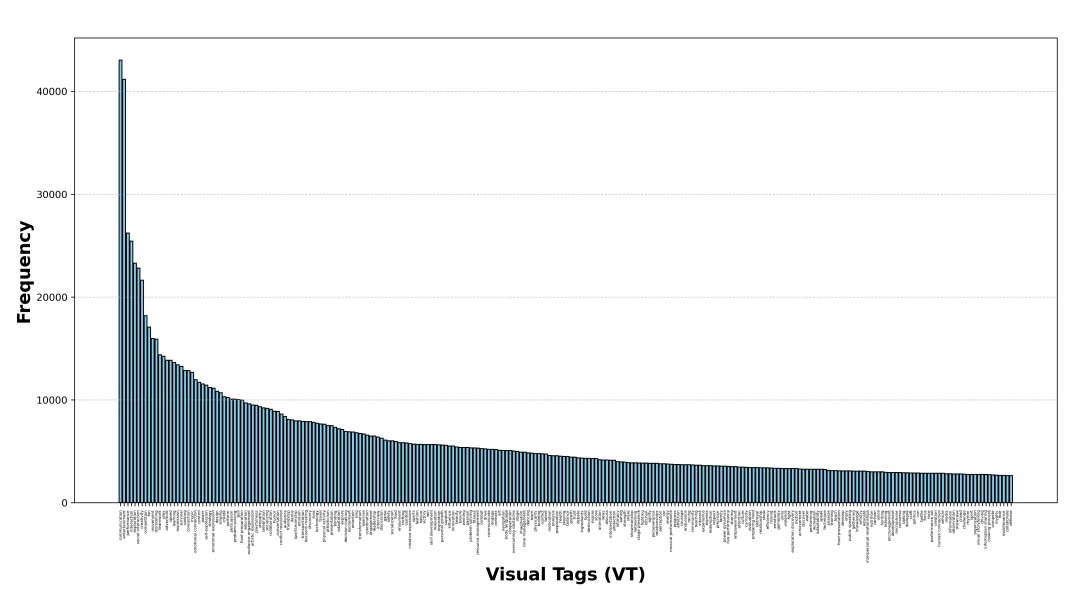

Figure 18: Distribution of the top-250 training visual tags of MSR-VTT.

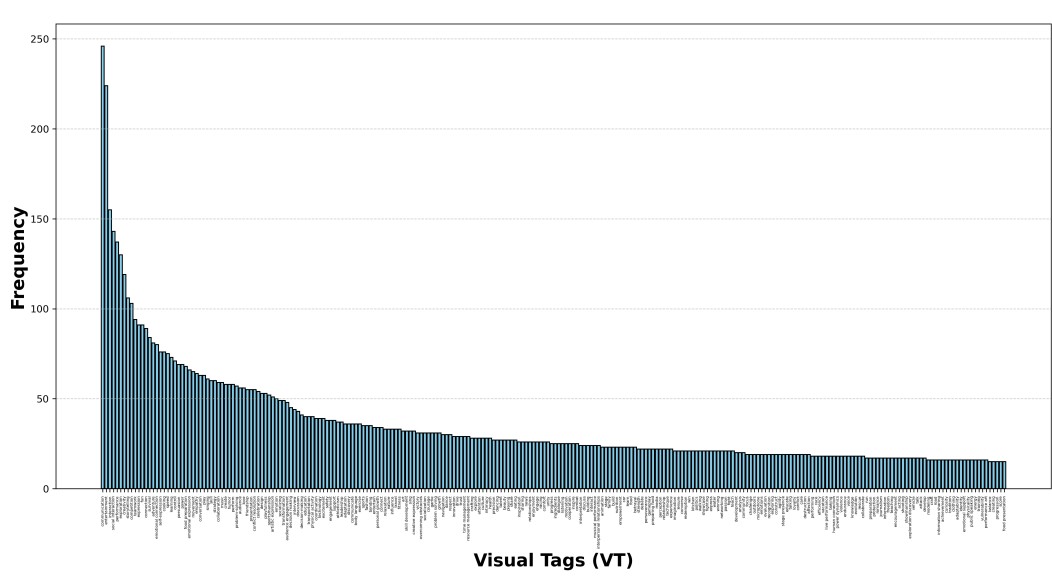

Figure 19: Distribution of the top-250 testing visual tags of MSR-VTT.

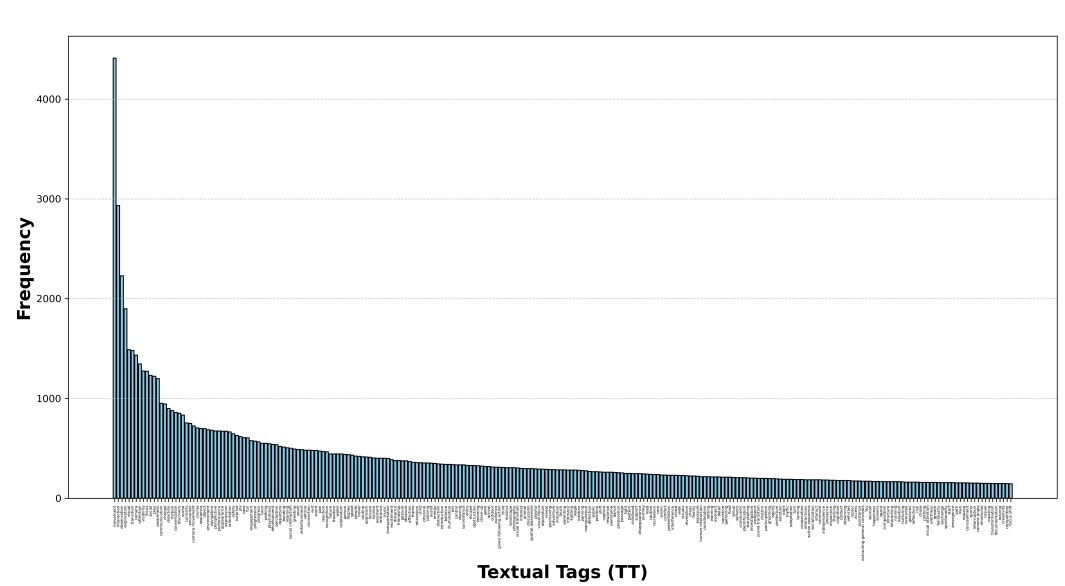

Figure 20: Distribution of the top-250 training textual tags of DiDeMo.

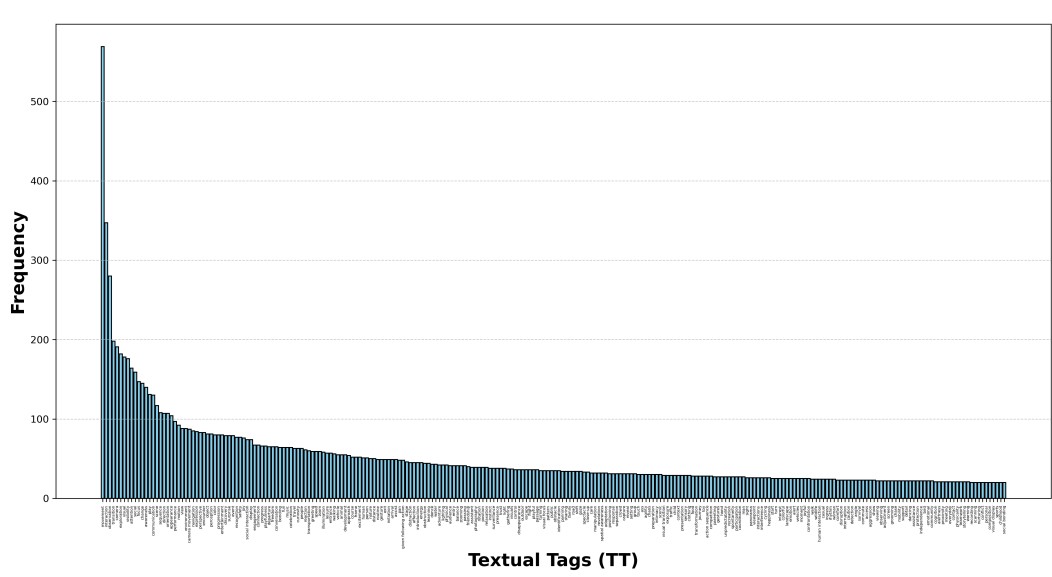

Figure 21: Distribution of the top-250 validation textual tags of DiDeMo.

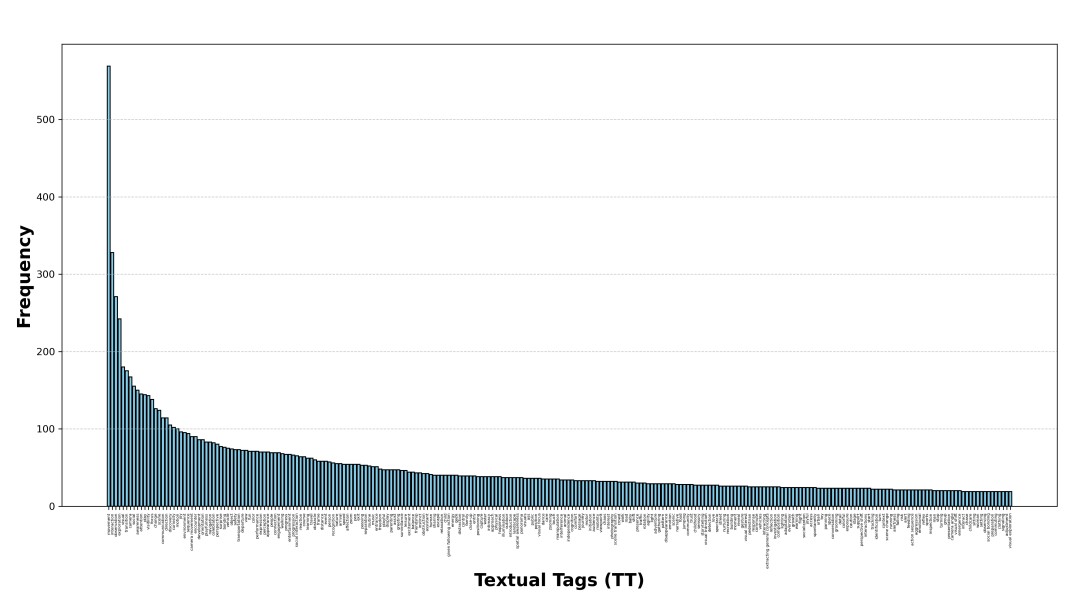

Figure 22: Distribution of the top-250 testing textual tags of DiDeMo.

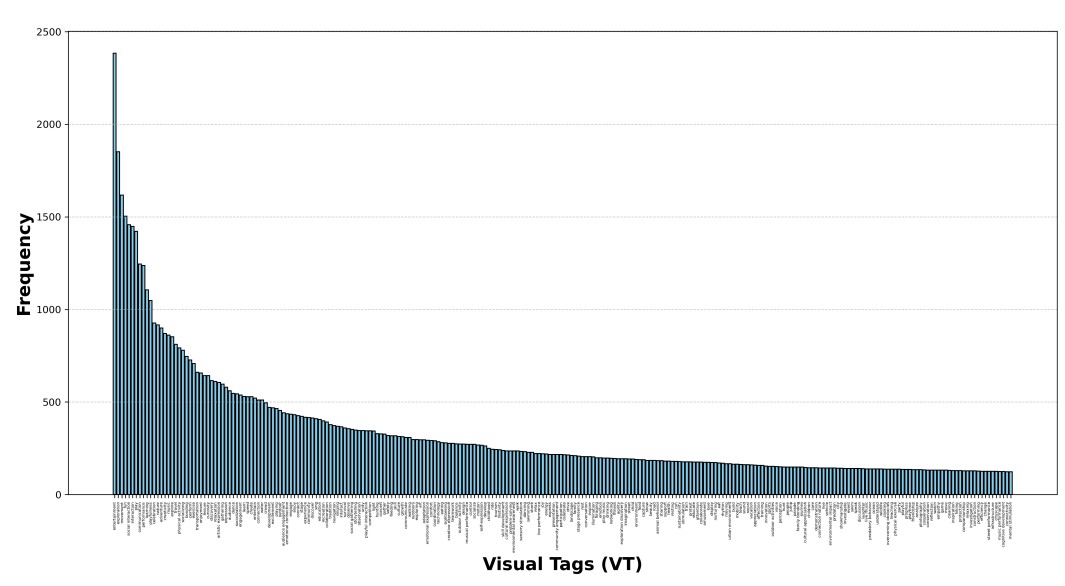

Figure 23: Distribution of the top-250 training visual tags of DiDeMo.

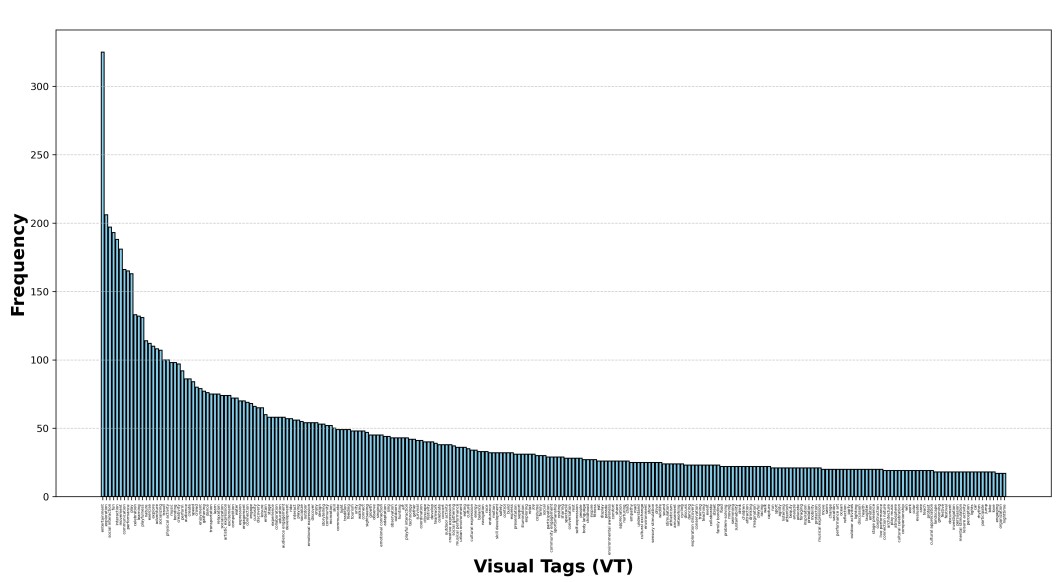

Figure 24: Distribution of the top-250 validation visual tags of DiDeMo.

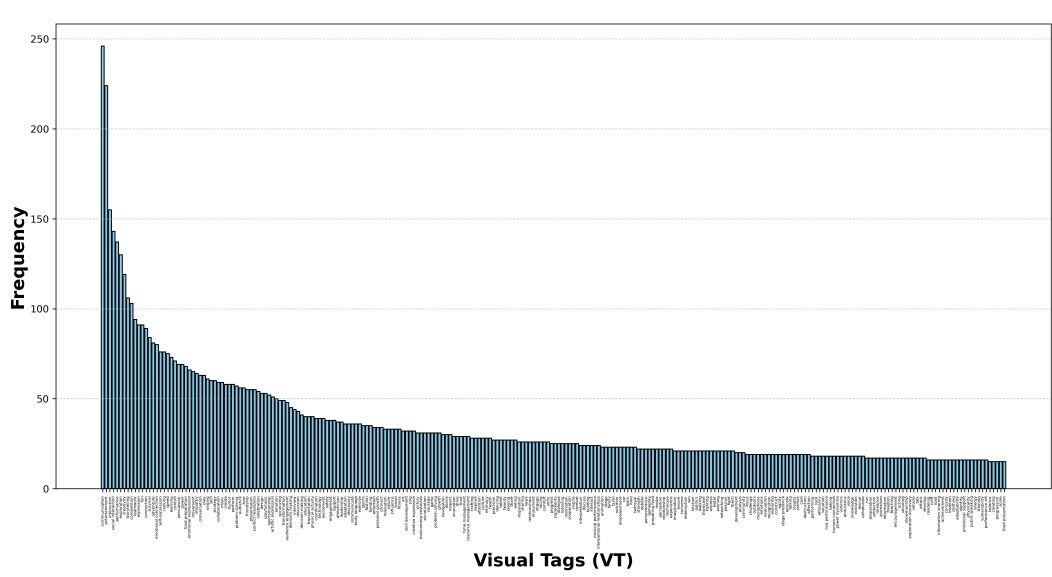

Figure 25: Distribution of the top-250 testing visual tags of DiDeMo.

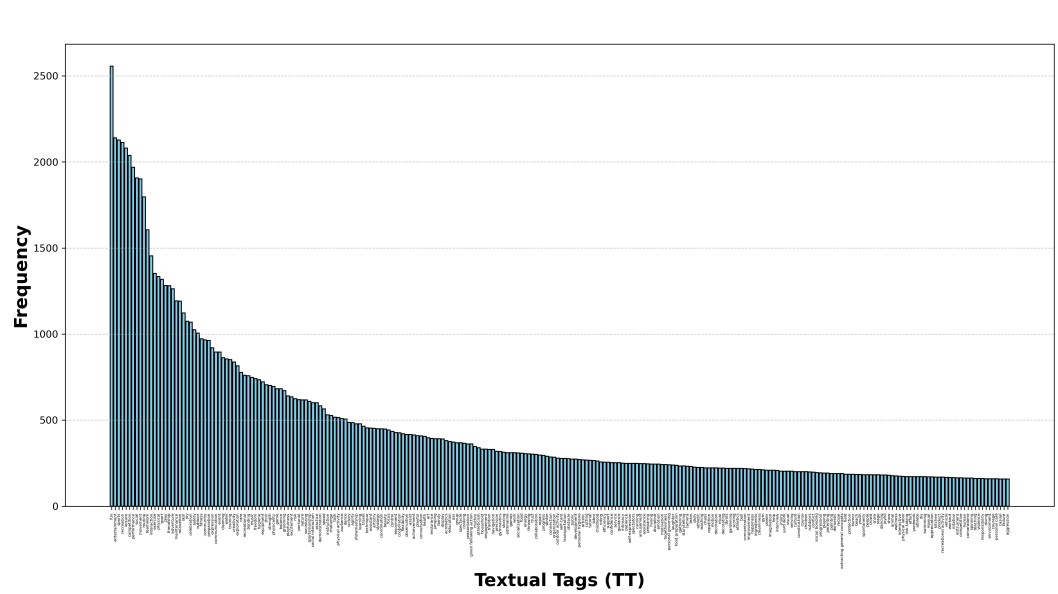

Figure 26: Distribution of the top-250 training textual tags of ActivityNet Captions.

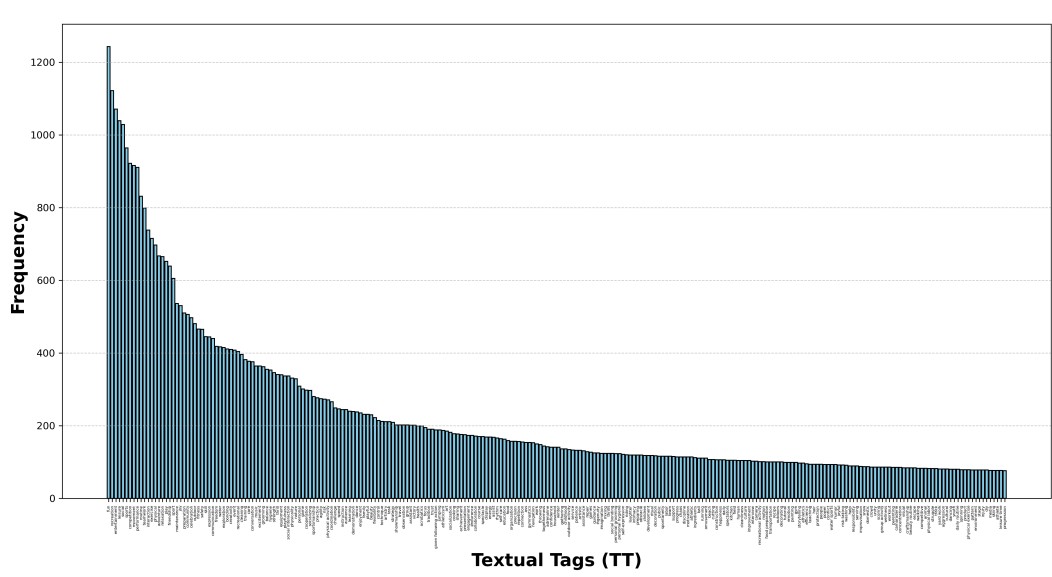

Figure 27: Distribution of the top-250 testing textual tags of ActivityNet Captions.

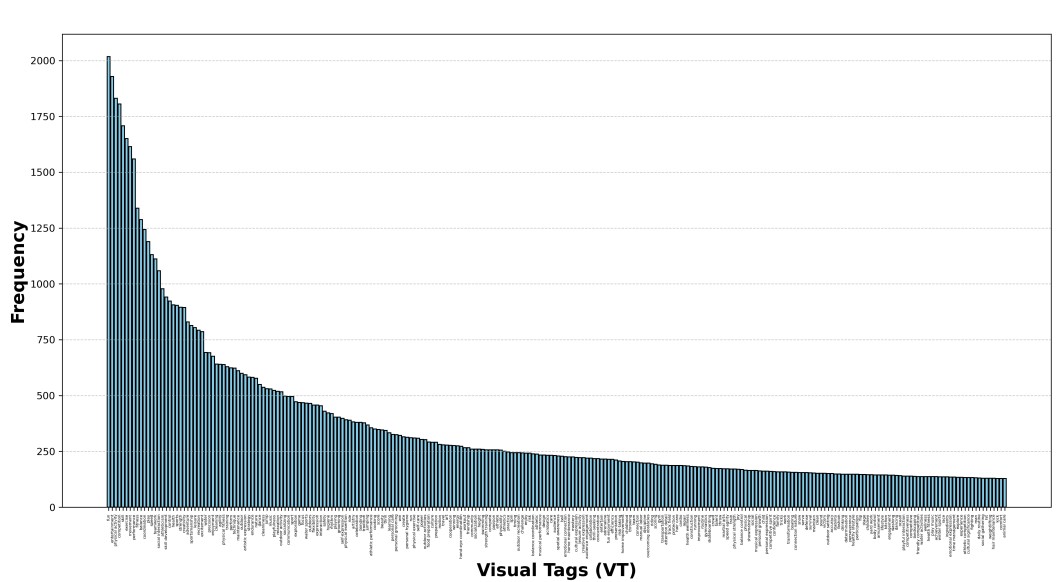

Figure 28: Distribution of the top-250 training visual tags of ActivityNet Captions.

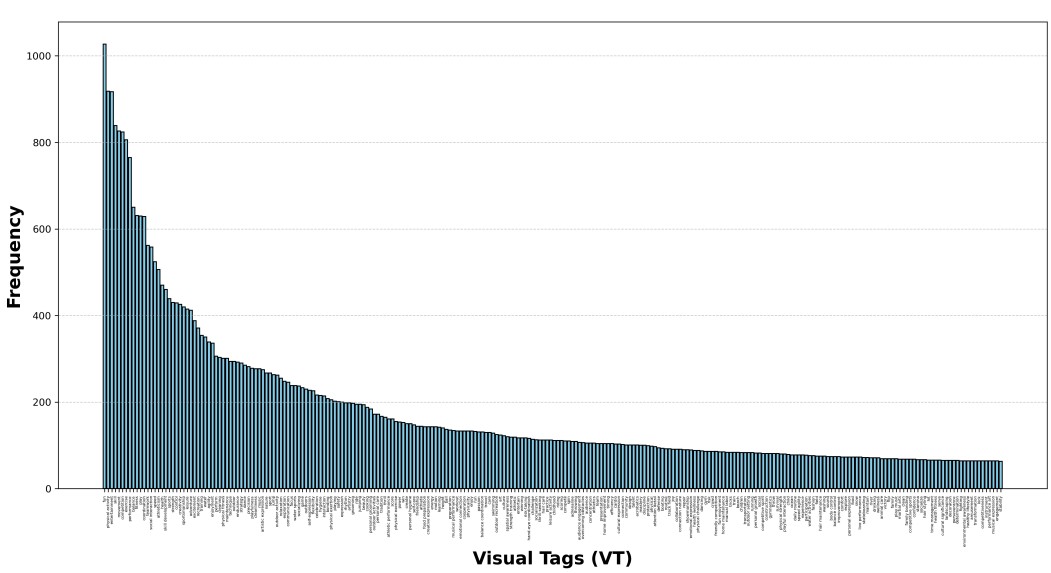

Figure 29: Distribution of the top-250 testing visual tags of ActivityNet Captions.

