# OpenReview forum: "Leveraging Modality Tags for Enhanced Cross-Modal Video Retrieval"
_ICLR.cc/2025/Conference — Submitted to ICLR 2025_

### Official Review · Reviewer_PUNk · 2024-10-24

**Soundness:** 3
**Presentation:** 2
**Contribution:** 2
**Rating:** 5
**Confidence:** 4

**Summary:**

This paper utilizes auxiliary concepts to improve the alignment of visual and textual latent concepts, enabling the distinction between each concept. It also introduces an Alignment Loss to strengthen the alignment between visual and textual latent concepts. Experimental results demonstrate the effectiveness of the proposed method.

**Strengths:**

This paper conducted thorough experiments to demonstrate the effectiveness of the method.

**Weaknesses:**

1. The paper contains numerous typographical errors; please review it carefully. For example, on line 142, it reads "into a shared ddimensional latent embedding"; on line 236, it says "Alignment Loss LA and"; on line 314, it mentions "Flikr videos"; and on line 317, it states "on the vall split". Figure 3 has low resolution. There are also missing bold elements in Table 2. The authors are requested to carefully proofread their paper. The spelling in Figure 5 also contains errors, such as "enjoyoing". The capitalization of the table labels throughout the paper is not consistent.
2.The LA loss proposed by the author is not clearly stated in the paper, and it seems to lack innovation.
3. The method proposed by the authors leverages the capabilities of VLM and LLM to generate tags for auxiliary alignment. However, as the authors' visualization results and experimental data analysis suggest, this kind of annotation largely generates confusing information that is detrimental to semantic alignment. This does not sound reasonable. Moreover, the authors still heavily rely on the alignment capabilities of the latent space itself, and the experiments do not reflect the effect of using only tags. I speculate that the effect obtained by using only tags is not good. The authors could further demonstrate the rationality and effectiveness of the proposed method through additional experiments or other means. It is difficult to confirm the effectiveness of the method from the current experimental data because I have observed that some results on DiDeMo and ActivityNet Captions have actually declined. This may be due to overfitting to the first dataset, rather than an improvement brought about by the method itself.

**Questions:**

none

---

> ### Author Response · Authors · 2024-11-22
> **Weakness 1**
>
> R4: "The paper contains numerous typographical errors":
>
> We thank the reviewer for highlighting the typographical errors. We have read the paper again carefully and have corrected those we have spotted in the new version of the paper. Figure 3 has been uploaded with high resolution and all the tables have been checked to be consistent with missing bold elements.
>
> R4: "on line 142, it reads "into a shared ddimensional latent embedding":
>
> L140 of the new version of the paper shows "d-dimensional latent embedding space" because as defined in L139 "$f_v : v_i \longrightarrow \Omega \in \mathbb{R}^d$" where $\Omega$ has dimensionality $d$.
>
> R4: "The capitalization of the table labels throughout the paper is not consistent":
>
> Sorry, we were unsure as to what table labels are not consistently capitalised, could you expand more so that we can fix this?
>
> R4: "The LA loss proposed by the author is not clearly stated in the paper, and it seems to lack innovation.":
>
> Please refer to comment 1) of reviewer JqAZ for details of the proposed Alignment loss $\mathcal{L_A}$. The novelty of this loss is not its formulation, which is the same one of [1] as we stated in L264-L267, but how we use it to better align the visual and textual concepts that come from a video and its caption. We propose to use auxiliary concepts that come from the generated visual and textual tags to better align the visual and textual concepts (Fig. 6 of the main paper already shows the better alignment of the visual and textual concepts compared to the original DiCoSA [1]) by the introduction of the Alignment loss $\mathcal{L_A}$. To better show this, we plot the t-SNE plot of visual and textual concepts on the MSR-VTT test set of MAC-VR with and without using $\mathcal{L_A}$ (in other words when we only generate auxiliary concepts and concatenate them with the visual and textual tags, see Fig.2). We included this visualization in Appendix H (see Fig. 10) of the new version of the paper. As can be seen, the use of $L_{A}$ helps to better distinguish the different concepts and have better clusters in the t-SNE plot.
>
> [1] Peng Jin, Hao Li, Zesen Cheng, Jinfa Huang, Zhennan Wang, Li Yuan, Chang Liu, and Jie Chen. Text-video retrieval with disentangled conceptualization and set-to-set alignment. In International Joint Conference on Artificial Intelligence (IJCAI), 2023b.

---

> ### Author Response · Authors · 2024-11-22
> **Weakness 2 Part.1**
>
> R4: "as the authors' visualization results and experimental data analysis suggest, this kind of annotation largely generates confusing information that is detrimental to semantic alignment":
>
> We are unsure as to what analysis and visualisation shows that the annotation generates confusing information, could you kindly provide specific details?
> Tab.2 of the main paper shows that we outperform the original DiCoSA[1] baseline across all the datasets (our model is based on the original DiCoSA architecture) and Fig. 5 shows examples of the generated tags on all the datasets showcasing the obtained improvement against our baseline and that the generated tags are related to the video and text from where they have been generated.
> Tab.3 shows that when using QB and DSL, our R@1 on MSR-VTT is the highest among all the work, and we achieve comparable results on DiDemo and ActivityNet.
> As we stated in L412-L414, many SOTA works use different training parameters when training on DiDemo and ActivityNet, more precisely different values of Batch Size ($BS$) and Number of Frames ($N_v$), such as EMCL-Net [2] and UCoFIA[3]. We re-run UCoFIA and EMCL-Net by using our same training parameters:
>
> | Dataset   | Method      | IS  | R@1  | R@5  | R@10 | MR | MeanR |
> |-----------|-------------|-----|-------|-------|------|----|-------|
> | DiDeMo    | UCoFIA[3]   |  -  | 42.1  | 71.7  | 80.3 | 2  | 16.3  |
> | DiDeMo    | MAC-VR      |  -  | 43.4  | 72.7  | 82.3 | 2  | 16.9  |
> | DiDeMo    | EMCL-Net[2] | DSL | 47.6  | 73.5  | 82.8 | 2  | 11.9  |
> | DiDeMo    | MAC-VR      | DSL | 50.2  | 76.2  | 84.2 | 1  | 15.1  |
>
> | Dataset      | Method      | IS  | R@1  | R@5  | R@10 | MR | MeanR |
> |--------------|-------------|-----|-------|-------|------|----|-------|
> | ActivityNet  | UCoFIA[3]   |  -  | 41.2  | 73.6  | 84.3 | 2  |  7.9  |
> | ActivityNet  | MAC-VR      |  -  | 37.9  | 69.4  | 81.5 | 2  |  9.6  |
> | ActivityNet  | EMCL-Net[2] | DSL | 47.1  | 75.7  | 86.4 | 2  |  7.0  |
> | ActivityNet  | MAC-VR      | DSL | 46.5  | 75.6  | 86.2 | 2  |  6.9  |
>
>
> MAC-VR outperforms UCoFIA on DiDeMo and using our same training parameters the performance on ActivityNet of UCoFIA drops drastically getting closer performance to MAC-VR. Similarly, MAC-VR outperforms EMCL-NeT on DIDeMo and gets very similar performance on ActivityNet, see Tab.9 of the appendix of the new version of the paper.
>
> We compared the importance of tags against the use of generated captions from foundation models in Tab.7 of the main paper to show the effectiveness of using tags instead of new captions. Additionally, as we stated in L400-L402 we outperform TABLE[4] when using DSL on MSR-VTT and DIDeMo. This method is similar to ours in that it extracts visual tags by using pre-trained experts and audio information. However, we use only visual and textual information to extract tags by using foundation models.\\
>
>
> [1] Peng Jin, Hao Li, Zesen Cheng, Jinfa Huang, Zhennan Wang, Li Yuan, Chang Liu, and Jie Chen. Text-video retrieval with disentangled conceptualization and set-to-set alignment. In International Joint Conference on Artificial Intelligence (IJCAI), 2023b.
>
> [2] Peng Jin, Jinfa Huang, Fenglin Liu, Xian Wu, Shen Ge, Guoli Song, David A. Clifton, and Jie Chen. Expectation-maximization contrastive learning for compact video-and-language representations. In Conference on Neural Information Processing Systems (NeurIPS), 2022.
>
> [3] Ziyang Wang, Yi-Lin Sung, Feng Cheng, Gedas Bertasius, and Mohit Bansal. Unified coarse-to-fine alignment for video-text retrieval. In International Conference on Computer Vision (ICCV),2023b.
>
> [4] Yizhen Chen, Jie Wang, Lijian Lin, Zhongang Qi, Jin Ma, and Ying Shan. Tagging before alignment: Integrating multi-modal tags for video-text retrieval. In Conference on Artificial Intelligence (AAAI), 2023.

---

> ### Author Response · Authors · 2024-11-22
> **Weakness 2 Part.2**
>
> R4: "and the experiments do not reflect the effect of using only tags. I speculate that the effect obtained by using only tags is not good":
>
> The main intuition of the paper is not to find a substitution of the original caption as a query but to show that using textual tags generated (both from video and text) we can better align the two different modalities (as shown in Fig.6 of the main paper) and boost the results compared to our baseline (see Tab.2). Using only tags would cause performance to drop as they are not meant to replace the caption/video to discriminate between videos/captions.
> Our results consistently show that tags offer an additional benefit for video retrieval, providing complementary information to the caption/video.
>
> R4:"The authors could further demonstrate the rationality and effectiveness of the proposed method through additional experiments or other means. It is difficult to confirm the effectiveness of the method from the current experimental data"
> :
> We find this comment unspecific and unclear as you mention one of the strengths of our paper is: "This paper conducted thorough experiments to demonstrate the effectiveness of the method". This contradicts this weakness.
>
> R4:"overfitting to the first dataset, rather than an improvement brought about by the method itself.": We do not find any evidence of overfitting on MSR-VTT, our performance across all datasets when using tags improves results compared to our baseline DiCoSA (see Tab.2 of the main paper) and better align the visual and textual modality (see Fig.6 of the main paper).

---

> ### Author Response · Authors · 2024-11-25
>
> We are still looking forward to hear from Reviewer PUNk before the discussion deadline.
>
> We believe we have addressed the raised concerns and summarise our responses below:
> - We answered Weakness 1 by reading the paper again carefully and correcting all the typographical errors we have spotted. Figure 3 has been uploaded with high resolution and all the tables have been checked to be consistent with missing bold elements. We explained in detail our proposed Alignment Loss $\mathcal{L}_{A}$ and the novelty of this loss. We asked also more details on the capitalization problem of tables spotted by the reviewer.
> - We answered Weakness 2 by showing that the generated tags do not create confusing information, see Tab2  and Fig.5 of the main paper where we outperform our baseline in all the scenarios and datasets. Moreover we re-run some SOTA works using our same pre-training parameters to show that our method outperforms these on DiDemo and get similar results on ActivityNet. We explain why using only tags as query obviously drops the retrieval performance and show that the extracted tags offer an additional benefit for video retrieval without over-fitting on all the datasets, see Tab.2 and Tab.3 of the main paper.

---

> > ### Comment · Reviewer_PUNk · 2024-11-26
> >
> > Thanks for the reply of the authors, I decide to raise my score to "5".

---

### Official Review · Reviewer_JqAZ · 2024-10-25

**Soundness:** 3
**Presentation:** 3
**Contribution:** 3
**Rating:** 8
**Confidence:** 5

**Summary:**

This paper proposes a method called Modality Auxiliary Concepts to enhance video retrieval performance, utilizing large models to generate Visual/Textual Tags that help align visual and textual concepts. I hold a positive view of this research.

**Strengths:**

This paper proposes a method called Modality Auxiliary Concepts to enhance video retrieval performance, utilizing large models to generate Visual/Textual Tags that help align visual and textual concepts. The approach is also logically clear and well-structured.

**Weaknesses:**

1、In line 300, for the proposed alignment loss $L_{A}$, please provide the principle or the formula for further clarification.
2、2. I suggest formatting the tables in the style of three-line tables for improved aesthetics. In line 472, $L_{L_{A}}$ should be changed to $L_{A}$.

**Questions:**

1、1. Please refer to Figure 3 and answer this question: During model testing, are the features $T_{i}$ generated by the Text encoder used as input to the T-CVE model?
2、In line 362, it is mentioned that K=8. I believe that the choice of K is crucial, and I recommend an ablation study to investigate the impact of different values of K.

---

> ### Author Response · Authors · 2024-11-22
> **Weakness 1**
>
> R3: "In line 300, for the proposed alignment loss $L_{A}$, please provide the principle or the formula for further clarification.":
>
> We have made the definition of the proposed alignment loss $\mathcal{L_A}$ clearer in Appx.D of the new version of the paper. As mentioned in L264-L267, our alignment loss $\mathcal{L_A}$ follows the same formulation of the contrastive loss $\mathcal{L_C}$ proposed in [1]. The $\mathcal{L_C}$ loss consists of two terms: an Inter-Concept Decoupling and an Intra-Concept Alignment term.
> The Inter-Concept Decoupling loss aims to ensure that latent subspaces capturing different semantic aspects of text and video representations are minimally correlated. This separation allows each subspace to focus on unique semantic features without overlap, enhancing the overall discriminative power. Mutual information between latent factors is used to quantify their dependency. However, since direct computation is challenging, the covariance $C_{k,l}$ between normalized latent factors is used as a proxy. By minimizing this covariance for unrelated subspaces through a loss function:
> \begin{equation}
> L_1=\sum_k\sum_{l\neq k}(C_{k,l})^2
> \end{equation}
> the model effectively reduces inter-concept mutual dependencies, achieving conceptual disentanglement.
> The Intra-Concept Alignment loss focuses on strengthening the correspondence between text and video representations within the same semantic subspaces. By maximizing mutual information between positive pairs, the alignment ensures that corresponding subspaces align semantically. This is implemented through a loss:
>
> \begin{equation}
> L_2=\sum_k(1-C_{k,k})^2
> \end{equation}
>
> which encourages high covariance for aligned pairs, ensuring that the semantic alignment within subspaces is robust and accurate.
> The combination of these two approaches is captured in the total loss function
>
> \begin{equation}
> \mathcal{L_C}=\gamma L1+\delta L_2
> \end{equation}
>
> where weights balance the importance of decoupling and alignment. Together, these mechanisms isolate distinct semantic concepts while aligning corresponding text and video representations, enabling efficient and accurate cross-modal retrieval.
> The weights $\gamma$ and $\delta$ have been ablated in [1].
> Our Alignment Loss $\mathcal{L_A}$ has the same formulation as explained above and we use the same weights already ablated in [1]. The difference with $\mathcal{L_C}$ is that "for each disentangled concept pair $(e_{i,k}^v, e_{i,k}^{a^v})$ we minimise the distance between this pair then maximise the distance to other disentangled concepts, i.e. $(e_{i,k}^v, e_{i,l}^{a^v}); l \ne k$ in a contrastive fashion to align modality concepts, similar to the loss in [1]." Similarly, we do the same with the pair $(e_{i,k}^t, e_{i,k}^{a^t})$ and $(e_{i,k}^t, e_{i,l}^{a^t}); l \ne k$.
>
> [1] Peng Jin, Hao Li, Zesen Cheng, Jinfa Huang, Zhennan Wang, Li Yuan, Chang Liu, and Jie Chen. Text-video retrieval with disentangled conceptualization and set-to-set alignment. In International Joint Conference on Artificial Intelligence (IJCAI), 2023b.

---

> ### Author Response · Authors · 2024-11-22
> **Weakness 2**
>
> R3: "In line 472, $L_{LA}$ should be changed to $L_{A}$":
>
> Thank you for spotting this typo, we have corrected this in the newly uploaded version of the paper.

---

> ### Author Response · Authors · 2024-11-22
> **Question 2**
>
> R3: "In line 362, it is mentioned that K=8. I believe that the choice of K is crucial, and I recommend an ablation study":
>
> As we stated in L322: "We use the base code from DiCoSA [1] for our architecture". We used the  original implementation of DiCoSA without changing any hyper-parameters that have already been ablated in the original architecture. In [1], the authors tested the original DiCoSA architecture with different values of $K\in\{2, 4, 8, 16, 32\}$ and they got the best performance with $K=8$. To verify [1]'s findings, we run the ablation when changing the value of K for MSR-VTT. We arrive at the same conclusion of the original DiCoSA [1] as shown from the following table:
>
> | Dataset   | K   | R@1  | R@5  | R@10 | MR | MeanR |
> |-----------|-----|-------|-------|------|----|-------|
> | MSR-VTT   | 4   | 52.4  | 77.1  | 85.1 | 1  | 10.2  |
> | MSR-VTT   | 8   | 53.2  | 77.7  | 85.3 | 1  | 10.0  |
> | MSR-VTT   | 16  | 52.0  | 77.3  | 84.8 | 1  | 10.4  |
> | MSR-VTT   | 32  | 50.7  | 77.1  | 84.5 | 1  | 10.3  |
>
>
> [1] Peng Jin, Hao Li, Zesen Cheng, Jinfa Huang, Zhennan Wang, Li Yuan, Chang Liu, and Jie Chen. Text-video retrieval with disentangled conceptualization and set-to-set alignment. In International Joint Conference on Artificial Intelligence (IJCAI), 2023b.

---

> ### Author Response · Authors · 2024-11-23
> **Question 1**
>
> R3: "During model testing, are the features $T_i$ generated by the Text encoder used as input to the T-CVE model":
>
> Yes, also during inference the text features $T_i$ are used as input to the T-CVE model. Note, we ensure that no information is leaked at test time by extracting tags for every video in the gallery -- not only the ground truth video, as we stated in L196-197.

---

> ### Comment · Reviewer_JqAZ · 2024-11-24
>
> I have no other suggestions for the paper and have decided to accept it.

---

> > ### Author Response · Authors · 2024-11-25
> >
> > We thank the reviewer for their positive decision to accept the paper, their time reading our work, and invaluable comments given regarding our paper.

---

### Official Review · Reviewer_UWvh · 2024-11-02

**Soundness:** 2
**Presentation:** 1
**Contribution:** 2
**Rating:** 3
**Confidence:** 5

**Summary:**

This paper introduces Modality Auxiliary Concepts for video retrieval, a novel approach that leverages modality-specific tags to enhance video retrieval. However, the novelty of the paper is insufficient. It only uses a large model to obtain the tag information of text and video to enhance the performance of the model. In addition, based on the introduction of additional data and large model knowledge, our model still lags far behind the state-of-the-art methods.

**Strengths:**

1. This paper introduces Modality Auxiliary Concepts for video retrieval, a novel approach that leverages modality-specific tags to enhance video retrieval
2. The authors propose to extract modality-specific tags from foundational VLMs and LLMs to augment the video and text modalities.
3. The authors propose a new Alignment Loss to better align and distinguish these learnt latent concepts.

**Weaknesses:**

1. The novelty of the paper is insufficient. It only uses LLMs to obtain the tag information of text and video to enhance the performance of the model. In addition, based on the introduction of additional data and large model knowledge, our model still lags far behind the state-of-the-art methods including T-MASS as shown in Table 3.
2. The performance of the proposed method in this paper lags significantly behind the state-of-the-art (SOTA) techniques. Upon examining Table 3, it is evident that the proposed MAC-VR method falls behind T-MASS in terms of R@5 and R@10 for the MSRVTT dataset. For DiDeMo, MAC-VR also underperforms T-MASS across R@1, R@5, R@10, and MeanR metrics. The disparity in performance is notably substantial. Also, for ActivityNet, MAC-VR also falls behind T-MASS at R@1 and R@5 by a large margin.
3. The logic in L53 is strange. The authors explain the effectiveness of different inference strategies which seems to have nothing to do with the auxiliary concept loss function described later.
4. Some important papers need to be referenced and compared, including:
[1] Clip-vip: Adapting pretrained image-text model to video-language representation alignment. ICLR 2023
[2] Uatvr: Uncertainty-adaptive text-video retrieval. ICCV 2023
5. The writing of the paper is not professional and needs further improvement. The symbols of this paper should be consistent, such as QB in L157 vs \textit{QB} in L158, maybe $\in$in L140,   et al.
6. The second term in Eq. 5 should also be preceded by a weight parameter to control it, and the selection of its parameters should be verified through experiments.
7. The authors do not define the K in Eq. 4. Besides, the K in Eq. 4 seems to be different from the K in L323, so I suggest the authors use different symbols to distinguish them.

**Questions:**

1. The novelty of the paper is insufficient. It only uses LLMs to obtain the tag information of text and video to enhance the performance of the model. In addition, based on the introduction of additional data and large model knowledge, our model still lags far behind the state-of-the-art methods including T-MASS as shown in Table 3.
2. The performance of the proposed method in this paper lags significantly behind the state-of-the-art (SOTA) techniques. Upon examining Table 3, it is evident that the proposed MAC-VR method falls behind T-MASS in terms of R@5 and R@10 for the MSRVTT dataset. For DiDeMo, MAC-VR also underperforms T-MASS across R@1, R@5, R@10, and MeanR metrics. The disparity in performance is notably substantial. Also, for ActivityNet, MAC-VR also falls behind T-MASS at R@1 and R@5 by a large margin.
3. The logic in L53 is strange. The authors explain the effectiveness of different inference strategies which seems to have nothing to do with the auxiliary concept loss function described later.
4. Some important papers need to be referenced and compared, including:
[1] Clip-vip: Adapting pretrained image-text model to video-language representation alignment. ICLR 2023
[2] Uatvr: Uncertainty-adaptive text-video retrieval. ICCV 2023
5. The writing of the paper is not professional and needs further improvement. The symbols of this paper should be consistent, such as QB in L157 vs \textit{QB} in L158, maybe $\in$in L140,   et al.
6. The second term in Eq. 5 should also be preceded by a weight parameter to control it, and the selection of its parameters should be verified through experiments.
7. The authors do not define the K in Eq. 4. Besides, the K in Eq. 4 seems to be different from the K in L323, so I suggest the authors use different symbols to distinguish them.

---

> ### Author Response · Authors · 2024-11-22
> **Weaknesses 1-2**
>
> R2: "The novelty of the paper is insufficient. It only uses LLMs to obtain the tag information of text and video to enhance the performance of the model.":
>
> The novelty of MAC-VR is not simply that it "uses LLMs to obtain the tag information of text and video to enhance the performance of the model", instead, we propose a novel approach to improve the alignment of visual and textual modalities by introducing tag information extracted respectively from videos and captions. Previous works, such as TABLE[1], propose a similar idea but focus only on tags extracted from the video and audio modality. We propose and show the importance of using also tags extracted from the text modality. We outperform these works such as TABLE, see Tab.3 of the main paper, and Fig 6 of the main paper shows that we are able to get better alignment between the different modalities when introducing visual and textual tags with our Alignment Loss $\mathcal{L}_{A}$.
>
> R2: "The performance of the proposed method in this paper lags significantly behind the state-of-the-art (SOTA) techniques.":
>
> Looking at Tab. 3, we can say that we outperform all SOTA on MSR-VTT when using QB and DSL as inference strategies and get comparable results without any inference strategies. A similar conclusion can be made for DiDeMo and ActivityNet. As we stated in L412-L414, many SOTA works use different training parameters on these two datasets, making the comparison not fair.
>
> R2: "our model still lags far behind the state-of-the-art methods including T-MASS as shown in Table 3.":\
> R2: "Upon examining Table 3, it is evident that the proposed MAC-VR method falls behind T-MASS ":
>
> The main idea of T-MASS is to treat a text as a "mass" in the embedding space rather than a single point, enabling a semantic range that accounts for potential misalignment between video and text embedding.
> This idea has been applied both during training and inference.
> Specifically, the inference pipeline has been reformulated to fully exploit the text mass for more effective text-video retrieval.
> Specifically, for each video candidate, T-MASS samples multiple stochastic text embeddings for the query text and select the closest one to the video embedding for the evaluation, using 20 sampling trials.
>
> Therefore, the results obtained by the T-MASS paper are not fairly comparable with our results due to a different inference process: T-MASS consider additional query text during testing whereas we consider only a single query text.
> Furthermore, a GitHub issue raised concerns about data leakage in T-MASS's inference pipeline, with several researchers identifying errors in the code:
> 1) Misaligned matrix dimensions in similarity computations lead to ground truth leakage, inflating results.
> 2) Simplifying the testing code significantly reduced results, confirming reliance on data leakage for inflated performance (e.g., R@1 dropped from 49.4 to 41.9 on MSR-VTT).
> 3) The support_loss negatively affected optimization and was inconsistently applied across datasets.
> 4) Complex testing functions with unnecessary constraints on video and text counts added further issues.
> 5) Overall, after removing the data leakage, the results were actually worse than the XPool results referenced in the main paper.
>
> The code of T-MASS has now been removed following this issue, this action, supported by other users, strongly suggests the presence of data leakage. Screenshots of the GitHub discussions are available for more detailed examination if required.
>
> R2: "Also, for ActivityNet, MAC-VR also falls behind T-MASS at R@1 and R@5 by a large margin.":
>
> T-MASS architecture has never been tested on ActivityNet dataset from the original authors.
>
> [1] Yizhen Chen, Jie Wang, Lijian Lin, Zhongang Qi, Jin Ma, and Ying Shan. Tagging before alignment: Integrating multi-modal tags for video-text retrieval. In Conference on Artificial Intelligence (AAAI), 2023.

---

> ### Author Response · Authors · 2024-11-22
> **Weakness 3**
>
> R2: "The logic in L53 is strange":
>
> Thank you for the suggestion, we have updated it in the new version of the paper (Lines 53-72) to make it clear that the two sentences are not related, rather that they are two different points to be made:
> "In this paper, we analyse the impact of such strategies on our MAC-VR architecture to ensure fair comparison with state-of-the-art (SOTA) methods. Our results show that auxiliary concepts of
> both modalities, in addition to the Alignment Loss, help boost the retrieval performance and better
> distinguish the latent concepts."

---

> ### Author Response · Authors · 2024-11-22
> **Weakness 4**
>
> R2: "Some important papers need to be referenced and compared":\
> Thanks to have spotted these two missing papers, we have added them to the related works and in the comparison with SOTA in the new version of the paper. Here is the comparison of MAC-VR against these two papers :
>
> | Dataset   | Method      | IS  | R@1  | R@5  | R@10 | MR | MeanR |
> |-----------|-------------|-----|-------|-------|------|----|-------|
> | MSR-VTT   | CLIP-ViP'[1]|  -  | 46.5  | 72.1  | 82.5 | -  |   -   |
> | MSR-VTT   | CLIP-ViP[1] |  -  | 50.1  | 74.8  | 84.6 | -  |   -   |
> | MSR-VTT   | UATVR[2]    |  -  | 47.5  | 73.9  | 83.5 | 2  | 12.3  |
> | MSR-VTT   | MAC-VR      |  -  | 48.8  | 74.4  | 83.7 | 2  | 12.3  |
> | MSR-VTT   | CLIP-ViP[1] | DSL | 55.9  | 77.0  | 86.8 | -  |   -   |
> | MSR-VTT   | UATVR[2]    | DSL | 49.8  | 76.1  | 85.5 | 2  | 12.3  |
> | MSR-VTT   | MAC-VR      | DSL | 53.2  | 77.7  | 85.3 | 1  | 10.0  |
>
> | Dataset   | Method      | R@1  | R@5  | R@10 | MR | MeanR |
> |-----------|-------------|-------|-------|------|----|-------|
> | DiDeMo    | CLIP-ViP'[1]| 40.6  | 70.4  | 79.3 | -  |   -   |
> | DiDeMo    | CLIP-ViP[1] | 48.6  | 77.1  | 84.4 | -  |   -   |
> | DiDeMo    | UATVR[2]    | 43.1  | 71.8  | 82.3 | 2  | 15.1  |
> | DiDeMo    | MAC-VR      | 43.4  | 72.7  | 82.3 | 2  | 16.9  |
> | DiDeMo    | CLIP-ViP[1] | 53.8  | 79.6  | 86.5 | -  |   -   |
> | DiDeMo    | MAC-VR      | 50.2  | 76.2  | 84.2 | 1  | 15.1  |
>
>  | Dataset   | Method      | R@1  | R@5  | R@10 | MR | MeanR |
> |-----------|-------------|-------|-------|------|----|-------|
> | ActivityNet | CLIP-ViP'[1]|   -   |   -   |   -  | -  |   -   |
> | ActivityNet | CLIP-ViP[1] | 51.1  | 78.4  | 88.3 | -  |   -   |
> | ActivityNet | MAC-VR      | 37.9  | 69.4  | 81.5 | 2  |  9.6  |
> | ActivityNet | CLIP-ViP[1] | 59.1  | 83.9  | 91.3 | -  |   -   |
> | ActivityNet | MAC-VR      | 46.5  | 75.6  | 86.2 | 2  |  6.9  |
>
> From the results we can conclude that MAC-VR performs better than UATVR [2] on both datasets. On the contrary, CLIP-ViP [1] outperforms MAC-VR on all the three datasets in both scenarios without inference strategy and when using DSL. Note that CLIP-ViP [1] adopts additional datasets as pre-training stage, e.g., WebVid-2.5M [3] and HD-VILA-100M [4] to better adapt CLIP. Therefore, the results against this model are not fairly comparable. Indeed, if we compare CLIP-ViP' [1], without pre-training, against MAC-VR, we outperform it on both datasets.
>
> [1] Hongwei Xue, Yuchong Sun, Bei Liu, Jianlong Fu, Ruihua Song, Houqiang Li, and Jiebo Luo. Clip-vip: Adapting pretrained image-text model to video-language representation alignment. In ICLR, 2023.
>
> [2] Bo Fang, Chang Liu, Yu Zhou, Min Yang, Yuxin Song, Fu Li, Weiping Wang, Xiangyang Ji, Wanli Ouyang, et al. Uatvr: Uncertainty-adaptive text-video retrieval. In ICCV, 2023.
>
> [3] Max Bain, Arsha Nagrani, G¨ul Varol, and Andrew Zisserman. A clip-hitchhiker’s guide to long video retrieval. arXiv,
> 2022.
>
> [4] Hongwei Xue, Tiankai Hang, Yanhong Zeng, Yuchong Sun, Bei Liu, Huan Yang, Jianlong Fu, and Baining Guo. Advancing high-resolution video-language representation with large-scale video transcriptions. In CVPR, 2022.

---

> ### Author Response · Authors · 2024-11-22
> **Weakness 5**
>
> R2: "The writing of the paper is not professional and needs further improvement":
>
> We thank the reviewer to have spotted inconsistent annotations in the main paper. We have read the paper again and have corrected all of them in the new version of the paper, if there are any remaining issues, we would happy to fix them.

---

> ### Author Response · Authors · 2024-11-22
> **Weakness 6**
>
> R2: "The second term in Eq. 5 should also be preceded by a weight parameter to control it, and the selection of its parameters should be verified through experiments.":
>
> As we stated in L296: "...the contrastive loss $\mathcal{L_C}$ to align the modality concepts as introduced in [1]" and in L322 "We use the base code from DiCoSA [1] for our architecture".
> We used the  original implementation of DiCoSA using the parameter values of $\mathcal{L_C}$ that have been already ablated by [1].
> We ablate the total weight of $\mathcal{L_A}$ because it was the additional proposed term of our training function.
> To verify [1]'s findings, we show the results ablation of the weight of $\mathcal{L_C}$:
>
> | Dataset   | $\eta$ | R@1  | R@5  | R@10 | MR | MeanR |
> |-----------|--------|-------|-------|------|----|-------|
> | MSR-VTT   | 0.5    | 52.5  | 76.5  | 85.6 | 1  | 11.0  |
> | MSR-VTT   | 1.0    | 53.2  | 77.7  | 85.3 | 1  | 10.0  |
> | MSR-VTT   | 2.0    | 52.6  | 77.2  | 85.8 | 1  | 10.5  |
> | MSR-VTT   | 5.0    | 52.0  | 76.8  | 85.3 | 1  | 11.1  |
>
>
> As shown from the results we get the best performance when using $\eta=1.0$ matching results in [1]. When using a larger $\eta$, the performance starts to drop, this is a proof of the fact that both the contrastive $\mathcal{L_C}$ and our alignment loss $\mathcal{L_A}$ must have equal weights during training to get the best performance. Moreover, this ablation also confirms that $\mathcal{L_A}$ helps the model to better align visual and textual concepts and so improve the final retrieval performance.
>
> [1] Peng Jin, Hao Li, Zesen Cheng, Jinfa Huang, Zhennan Wang, Li Yuan, Chang Liu, and Jie Chen. Text-video retrieval with disentangled conceptualization and set-to-set alignment. In International Joint Conference on Artificial Intelligence (IJCAI), 2023b.

---

> ### Author Response · Authors · 2024-11-22
> **Weakness 7**
>
> R2: "The authors do not define the $K$ in Eq. 4":
>
> We define and explain $K$ in L244-L245: "Inspired by[1], we disentangle each element of the quadruple into $K$ independent, equal-sized latent concepts. For example, when disentangling $V_i$, we get $K$ independent latent concepts, i.e. $E_i^v=[e^v_{i,1},..., e^v_{i,K}]$."
>
> R2: "Besides, the $K$ in Eq. 4 seems to be different from the $K$ in L323":
>
> The $K$ used in L323 is commonly used to represent the evaluation metric Recall at $K$. To avoid confusion, as suggested, we modify this $K$ with $L$ in line 320 of the new version of the paper (following [1]) and we define our loss symbol to $\mathcal{L}$ (not $L$) to ensure there is no confusion between these symbols.
>
> [1] Peng Jin, Hao Li, Zesen Cheng, Jinfa Huang, Zhennan Wang, Li Yuan, Chang Liu, and Jie Chen. Text-video retrieval with disentangled conceptualization and set-to-set alignment. In International Joint Conference on Artificial Intelligence (IJCAI), 2023b.

---

> ### Comment · Reviewer_UWvh · 2024-11-24
>
> The author has failed to address the concerns I previously raised. Primarily, the paper merely concentrates on incorporating tags of the text modality into video retrieval, which is an incremental work rather than a novel contribution suitable for publication in ICLR.
>
> For the performance evaluation, it is unacceptable that the author limits the comparisons within methods using some specific strategies (such as QB and DSL ). The methodology presented in the paper significantly lags behind the current state-of-the-art methods, including Clip-vip, T-MASS, etc.
>
> Furthermore, the author mentions data leakage issues in T-MASS and discrepancies between the reported performance and the actual results obtained from T-MASS. It is imperative to inquire whether the author has conducted thorough reproducibility tests on T-MASS's findings before reaching an unsubstantiated conclusion. Apart from T-MASS, the performance of the proposed method is also inferior to CLIP-vip.

---

> > ### Author Response · Authors · 2024-11-25
> > **Part 1**
> >
> > We struggle to understand why the reviewer states that their concerns have not been answered.
> > We already added new ablations and comparisons suggested by the reviewer that show the efficiency of the proposed methods.
> > We explain below and invite reviewer UWvh to be more explicit on where they believe the gap still exists. We are happy to add further clarifications before the deadline.
> >
> > We summarise how we already addressed the concerns again below:
> >
> > R2:" Primarily, the paper merely concentrates on incorporating tags of the text modality into video retrieval, which is an incremental work rather than a novel contribution":
> >
> > As we have already stated in the paper and in the public comments, our method does NOT "concentrate on incorporating tags of the text modality". We propose for the first time in video retrieval to use foundation models to extract tags not only from the video (as other papers have such as TABLE  using pre-trained experts and audio modality) but also from the text.
> > Importantly, we show that highlighting, and aligning, the tags from both modalities, improves performance significantly.
> > Our main premise is on the alignment of tags rather than just adding those tags.
> > We also proposed a new use of the contrastive loss (our Alignment Loss $\mathcal{L}_{A}$) to use the auxiliary tags to better align the video and text modality (see Tab. 4 and Fig. 10 for quantitative and qualitative results).
> >
> > We show from our qualitative results in Fig. 2 and in Sec. 3.2 of the main paper why using the text modality to extract tags: Tags extracted from the text modality are "abstract terms that go beyond the exact caption" (see L183-188), whereas tags extracted from the video modality enable us "to both capture a broader array of visual elements that characterise the video content and also have a representation of the video in words, facilitating matching the video content to the captions within the text modality." (see L189-194). Moreover, we show how these tags are helpful to better align the text and the video concepts in Fig. 6 of the main paper.
> >
> > R2: "For the performance evaluation, it is unacceptable that the author limits the comparisons within methods using some specific strategies (such as QB and DSL )."
> > We construct our results for direct and fair comparison for each previous work.
> > For each paper, we consider all results available in that paper and compare directly.
> > For example, if a method only provides results using QB inference strategy, then we fairly compare with our method using the same strategy.
> > We report our method's performance with and without these inference strategies.
> > We show that the choice of inference strategy can lead to large gains, independent of the method's actual contribution -- see Tab. 2 and Tab. 3.
> > Our paper is the first to dissect the inference strategy in prior works clarifying that these impact performance and need to be stated for direct comparison in each case.
> >
> > We believe we have fairly compared to every previous work. We ask the reviewer to identify what exactly do they wish us to compare? Because we indeed compare directly to every method using exactly the same strategy used in that work. This is a direct and fair comparison for each case.
> >
> > Under this fair comparison settings:
> > - We outperform all the works that use QB and DSL as inference strategies on MSR-VTT and we get comparable performance when not using any inference strategy.
> > - We reach similar conclusions on DiDeMo and ActivityNet. As we stated in response to Weaknesses 1-2 and in L412-L414 of the main paper, many SOTA works use different
> > training parameters (larger batch sizes during training/number of sampled frames) on these two datasets, making the comparison unfair.

---

> > > ### Comment · Reviewer_UWvh · 2024-11-26
> > >
> > > I have read the author's response, and I maintain my rating and confidence.

---

> > ### Author Response · Authors · 2024-11-25
> > **Part 2**
> >
> > R2: "The methodology presented in the paper significantly lags behind the current state-of-the-art methods, including Clip-vip, T-MASS, etc.": As we stated in response to Weaknesses 1,2 and 4 and in L402-L410 of the main paper:
> >
> > Clip-vip is NOT fairly comparable with MAC-VR because they do strong pre-training on WebVid-2.5M and HD-VILA-100M to better adapt CLIP and improve the retrieval performance. We show in response to Weakness 4 that we outperform Clip-vip on MSR-VTT and DiDemo **when there is no pre-training**.
> > We state in response to Weakness 1-2 and L405-410 of the main paper that T-MASS proposes a novel inference pipeline which uses **multiple textual queries at inference time**. We use **only a single text query** making the comparison in this situation not fairly comparable to ours. Importantly we highlight that this paper's results cannot be reproduced.
> > R2:" It is imperative to inquire whether the author has conducted thorough reproducibility tests on T-MASS's findings before reaching an unsubstantiated conclusion": As we stated in response of Weakness 1-2: "The code of T-MASS has now been removed following this issue (see the link here, this action, supported by other users (not only us). Multiple users have highlighted the same conclusions and the authors have failed to respond appropriately instead only withdrawing their code. This strongly suggests the presence of data leakage. Screenshots of the GitHub discussions are available for more detailed examination if required.
> >
> > R2: "Apart from T-MASS, the performance of the proposed method is also inferior to CLIP-vip." *As we already stated*, we outperform CLIP-Vip on MSR-VTT and DiDeMo when their strong pre-training is not used in Weakness 4. We again list the results here and are seriously concerned that our response is simply dismissed.
> >
> > | Dataset   | Method      | IS  | R@1  | R@5  | R@10 | MR | MeanR |
> > |-----------|-------------|-----|-------|-------|------|----|-------|
> > | MSR-VTT   | CLIP-ViP No-pretraining|  -  | 46.5  | 72.1  | 82.5 | -  |   -   |
> > | MSR-VTT   | CLIP-ViP |  -  | 50.1  | 74.8  | 84.6 | -  |   -   |
> > | MSR-VTT   | UATVR    |  -  | 47.5  | 73.9  | 83.5 | 2  | 12.3  |
> > | MSR-VTT   | MAC-VR      |  -  | 48.8  | 74.4  | 83.7 | 2  | 12.3  |
> > | MSR-VTT   | CLIP-ViP | DSL | 55.9  | 77.0  | 86.8 | -  |   -   |
> > | MSR-VTT   | UATVR    | DSL | 49.8  | 76.1  | 85.5 | 2  | 12.3  |
> > | MSR-VTT   | MAC-VR      | DSL | 53.2  | 77.7  | 85.3 | 1  | 10.0  |
> >
> > | Dataset   | Method      | R@1  | R@5  | R@10 | MR | MeanR |
> > |-----------|-------------|-------|-------|------|----|-------|
> > | DiDeMo    | CLIP-ViP No-pretraining| 40.6  | 70.4  | 79.3 | -  |   -   |
> > | DiDeMo    | CLIP-ViP | 48.6  | 77.1  | 84.4 | -  |   -   |
> > | DiDeMo    | UATVR[2]    | 43.1  | 71.8  | 82.3 | 2  | 15.1  |
> > | DiDeMo    | MAC-VR      | 43.4  | 72.7  | 82.3 | 2  | 16.9  |
> > | DiDeMo    | CLIP-ViP | 53.8  | 79.6  | 86.5 | -  |   -   |
> > | DiDeMo    | MAC-VR      | 50.2  | 76.2  | 84.2 | 1  | 15.1  |
> >
> >  | Dataset   | Method      | R@1  | R@5  | R@10 | MR | MeanR |
> > |-----------|-------------|-------|-------|------|----|-------|
> > | ActivityNet | CLIP-ViP No-pretraining|   -   |   -   |   -  | -  |   -   |
> > | ActivityNet | CLIP-ViP | 51.1  | 78.4  | 88.3 | -  |   -   |
> > | ActivityNet | MAC-VR      | 37.9  | 69.4  | 81.5 | 2  |  9.6  |
> > | ActivityNet | CLIP-ViP | 59.1  | 83.9  | 91.3 | -  |   -   |
> > | ActivityNet | MAC-VR      | 46.5  | 75.6  | 86.2 | 2  |  6.9  |

---

### Official Review · Reviewer_2ZNx · 2024-11-04

**Soundness:** 4
**Presentation:** 3
**Contribution:** 3
**Rating:** 8
**Confidence:** 3

**Summary:**

1. The authors in the paper introduce Modality Auxiliary Concepts for Video Retrieval, a novel approach that aims at improving alignment between video and text for effective video retrieval.
2. This work provides a framework for extracting and utilizing modality-specific tags using foundational models from video and text and an alignment loss is introduced to align modality-specific auxiliary concepts with visual and textual latent concepts.
3. In the paper they discuss how the architecture is tested with various inference strategies achieving competitive results across three benchmark datasets and ablation studies further validate the impact of each component.

**Strengths:**

1. The paper introduces an approach to use foundational models to extract modality-specific tags for both videos and text, this is a novel approach to enhance cross-model alignment.
2. Extensive experiments are conducted on three diverse datasets which cover a wide range of video retrieval tasks, comparison of MAC-VR against a strong baseline with SOTA methods and different inference strategies are documented.
3. The authors have documented extensive ablation studies for validating the contribution of each components such as the effect of different numbers of tags, foundation models and architectural choices on retrieval performance.

**Weaknesses:**

1. The paper does not cover the quality and the diversity of the tags extracted. An evaluation of tag relevance and coverage could enhance the contribution.
2. The paper has limited analysis of the individual contributions of video tags and text tags within the MAC-VR framework. Understanding the contribution of modality-specific tags to overall alignment can maybe help to identify the potential areas for improvement.

**Questions:**

1. Have the authors thought about ways to handle observed instances of hallucinated tags? And currently, how are these affecting retrieval accuracy?
2. Was a quantitative evaluation considered for the quality and relevance of extracted tags maybe like a comparison with human annotated tags?

Please also refer to weaknesses for other questions.

---

> ### Author Response · Authors · 2024-11-22
> **Weakness 1**
>
> R1: "The paper does not cover the quality and the diversity of the tags extracted. An evaluation of tag relevance and coverage could enhance the contribution.":
>
> Qualitative and Quantitative analysis of the extracted tags can be seen in Fig. 2, Tables 1 and 6, and Sec.3.2 of the main paper with more results in Figures 7, 8, 16-29 and Tables 10 and 11 in the appendix.
>
> In the main paper, Fig. 2 shows qualitative extracted visual and textual tags of the three considered datasets (see Sec.3.2 of the main paper for additional comments).
> Tab. 1 shows some statistical analysis of the extracted tags in all the considered datasets.
> Moreover, Tab.6 shows quantitative results when using different foundation models to extract visual and textual tags. From this table it is possible to confirm that the considered foundation models, i.e. VideoLLaMA2 and Llama3.1 generate better tags than Video-LLaMA and Llama2.
>
> In the appendix, we provide additional results and information on the quality and diversity of the extracted tags.
> Fig.7 shows additional examples of the extracted visual and textual tags across our datasets.
> It shows that the extracted tags are always relevant to the original video and caption.
> Fig.8 shows a qualitative comparison of the generated tags across our datasets when using different foundation models.
> It is evident that Video-LLaMA and Llama2 tend to hallucinate tags that are not relevant with what is  in the video/caption.
> Moreover, the textual tags extracted by Llama2 are very often words that already appear in the caption.
> On the contrary VideoLLaMA2 and Llama3.1 tend to extract tags that add additional information to the video and text (see Appendix F for more details and comments).
> A statistical analysis on the extracted tags when using different foundation models is done in Tab.10 and Tab.11.
> In L1129-L1133 we analysed the distribution of visual and textual tags generated across the datasets:"Fig.16 to Fig.29 shows the distribution of the top-250 visual and textual tags in training and testing of all the datasets. In general, we can see that the distribution of these tags is long-tailed and there are some tags that are very common. Consequently, the most common tags are shared among many pairs in the dataset, but we find that combinations of tags are still unique enough to provide discriminative information for the model."

---

> ### Author Response · Authors · 2024-11-22
> **Weakness 2**
>
> R1: "Understanding the contribution of modality-specific tags to overall alignment can maybe help to identify the potential areas for improvement":
>
> Thank you for the suggestion of this missing visualisation, we have added this in the new version of the appendix of our paper. We included these visualizations in Appendix H (see Fig.11 and Fig.12). The t-SNE plots illustrate that both visual and textual tags contribute to a clearer distinction between visual and textual concepts. However, visual tags demonstrate a stronger ability to achieve this differentiation. A possible explanation is that tags extracted from videos share the same modality as captions, which facilitates better alignment between visual and textual concepts.

---

> ### Author Response · Authors · 2024-11-22
> **Questions 1-2**
>
> R1: "Have the authors thought about ways to handle observed instances of hallucinated tags. And currently, how are these affecting retrieval accuracy?":
>
> We show the results of using different foundation models in Tab.6 of the main paper. Moreover, we showed examples and basic statistics when using different foundation models in the appendix (see Appendix F). We showed how Video-LLaMA and Llama2 tend to hallucinate more than the considered VideoLLaMA2 and Llama3.1. This is evident both from qualitative examples (See Fig.8 of Appendix F) and quantitative results (see Tab.6 of the main paper).
> Improving the foundational models used will reduce hallucinations. From qualitative analysis of our method, we don't believe hallucinated tags are causing issues with performance.
> Unfortunately, as far as we know there are no video retrieval datasets with human-labelled tags in the community to evaluate against.

---

> ### Author Response · Authors · 2024-11-25
>
> We thank the reviewer for supporting our paper's acceptance.
> We highlight that we have addressed the noted weaknesses, summarising below:
> - For Weakness 1, we highlighted that the Qualitative and Quantitative analysis is provided in the updated paper in Fig.2, Tables 1 and 6, and Sec.3.2 of the main paper with more results in Figures
> 7, 8, 16-29 and Tables 10 and 11 in the appendix.
> - We addressed Weakness 2 by plotting the t-SNE plots of visual and textual concepts when using only one modality-specific tag, see Fig.11 and Fig.12 in Appendix H.
> - We answered Questions 1-2 by showing qualitative and quantitative results when using a different foundation model to extract tags to handle observed instances of hallucinated tags.
>
> If there are additional questions, we are happy to answer them before the deadline.

---

> > ### Comment · Reviewer_2ZNx · 2024-11-26
> >
> > Thanks for the comments, I do not have any additional questions at this point.

---

### Meta-Review · Area_Chair_4Qf7 · 2024-12-12

**Metareview:**

The paper proposes latent concept tags for individual modalities for improving video retrieval. Reviewers praise the thorough experiments. However concerns are raised about novelty, method soundness, and some experimental/comparison choices. After the rebuttal, two reviewers respond that they maintain negative scores; these authors provided detailed reviews. Two reviewers maintain positive scores, but these reviews are less detailed. Overall the concerns do not seem sufficiently well addressed and support for the paper is countered by skepticism.

**Additional Comments On Reviewer Discussion:**

All reviewers participated in the discussion and responded as to whether their concerns were addressed (some increased their scores, some maintained them)

---

### Decision · Program_Chairs · 2025-01-22

Reject